# SYNC: Measuring and Advancing Synthesizability in Structure-Based Drug Design

**Yunfan Liu**[1,3,4,†], **Lirong Wu**[1,2,3,†], **Zhifeng Gao**[4], **Yufei Huang**[3], **Cheng Tan**[5],
**Haitao Lin**[3], **Zicheng Liu**[3], **Changxi Chi**[3], **Chang Yu**[3*], **Stan Z. Li**[3*]
[1]Zhejiang University      [2]DAMO Academy, Alibaba Group
[3]Westlake University      [4]DP Technology      [5]Shanghai AI Laboratory
{liuyunfan, wulirong, yuchang}@westlake.edu.cn

## Abstract

Designing 3D ligands that bind to a given protein pocket with high affinity is a fundamental task in Structure-Based Drug Design (SBDD). However, the lack of synthesizability of 3D ligands has been hindering progress toward experimental validation; moreover, computationally evaluating synthesizability is a non-trivial task. In this paper, we first benchmark eight classical synthesizability metrics across 11 SBDD methods. The comparison reveals significant inconsistencies between these metrics, making them impractical and inaccurate criteria for guiding SBDD methods toward synthesizable drug design. Therefore, we propose a simple yet effective SE(3)-invariant *SYNthesizability Classifier* (SYNC) to enable better synthesizability estimation in SBDD, which demonstrates superior generalizability and speed compared to existing metrics on five curated datasets. Finally, with SYNC as a plug-and-play module, we establish a synthesizability classifier-driven SBDD paradigm through guided diffusion and Direct Preference Optimization, where highly synthesizable molecules are directly generated without compromising binding affinity. Extensive experiments also demonstrate the effectiveness of SYNC and the advantage of our paradigm in synthesizable SBDD. Code is available at `https://github.com/XYxiyang/SYNC`.

## 1 Introduction

Recent years have witnessed the remarkable success of Artificial Intelligence-driven methods for drug design, of which Structure-Based Drug Design (SBDD) is a representative example (Guan et al., 2023; Peng et al., 2022; Luo et al., 2021). The primary objective of SBDD is to generate 3D ligands with desired properties that can bind tightly to a given protein pocket. Previous works typically formulate SBDD as a conditional generation task, where protein pockets are treated as conditions, and the distribution of 3D ligands is to be learned. Consequently, many deep generative models have been employed to handle this task, including autoregressive and diffusion-based models. For example, autoregressive models (Luo et al., 2021; Peng et al., 2022; Liu et al., 2022) generate atoms, bonds, or fragments of 3D ligands binding to a target pocket in a sequential manner. To overcome the dilemma of autoregressive models in error accumulation and generation order (Guan et al., 2024), diffusion-based models (Guan et al., 2023; Lin et al., 2025) directly learn the data distributions of atom types, atom position, and bond types from prior distributions. Although promising results in terms of binding-related metrics (e.g., docking scores) have been achieved, the synthesizability of the generated molecules, which is crucial for the practicality and validation of drug design, has continuously been an obstacle (Luo et al., 2024; Gao et al., 2024; Shen et al., 2025).

In practice, generating valid drugs with desirable properties from the vast chemical space (estimated to contain $10^{60}$ potential "drug-like" molecules) is extremely challenging, and considering synthesizability further exacerbates the difficulty. A central challenge here is the lack of well-enough criteria for evaluating the molecular synthesizability. There have been many efforts to evaluate synthesizability, which can be categorized into three branches. Rule-based methods (Ertl & Schuffenhauer, 2009; Voršilák et al., 2020) make qualitative predictions based on substructure, molecular mass, etc., which are efficient but fail to quantify synthesizability and lack generalizability.

---

†Equal contribution. *Corresponding authors.

Retrosynthetic-based methods (Genheden et al., 2020) guarantee the highest true positive rate since they output synthetic pathways, but are limited by the precursors in searching stock and are extremely time-consuming. Learning-based methods (Yu et al., 2022; Coley et al., 2018; Neeser et al., 2023) flexibly model molecules, but most utilize only coarse-grained features and fail to take 3D molecular conformations into account, which are needed by SBDD methods. We present the rankings of the 8 synthesizability metrics for the molecules generated by 11 SBDD methods in Figure 1. We observe huge inconsistencies between these metrics. Take the synthesizability of FLAG for example, 3 out of 8 metrics show poor synthesizability, while the others suggest good synthesizability. *These conflicting metrics and their respective drawbacks make it hard to draw a meaningful conclusion about the synthesizability and therefore limit the ability to guide synthesizable drug design.*

In this paper, we first establish a new benchmark for comprehensively comparing existing synthesizability metrics. Specifically, we have collected and organized 5 datasets containing both Easy- and Hard-to-Synthesize (ES/HS) molecules (Yu et al., 2022), as well as their 3D conformations, on which 8 classical synthesizability metrics are evaluated. We further propose a simple yet effective *SYNthesizability Classifier* (SYNC), which achieves the best performance in our benchmark and offers merits to guide SBDD methods. Together with existing metrics, we provide a comprehensive benchmark to evaluate the synthesizability of 11 SBDD methods. Finally, we provide two SYNC-driven paradigms that employ SYNC as a plug-and-play module to control the diffusion process: (i) guided diffusion with SYNC; (ii) DPO (Rafailov et al., 2024) with SYNC to achieve higher synthesizability while preserving binding affinity. Extensive experiments demonstrate

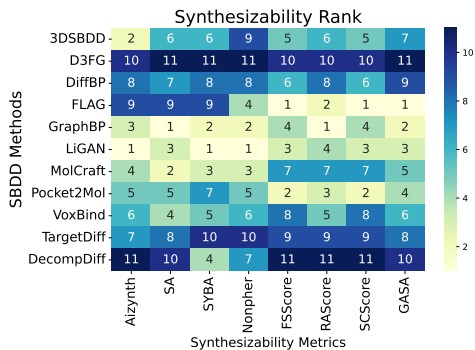

Figure 1: A comparison of rankings of 8 classical synthesizability metrics for molecules generated by 11 classical SBDD methods, which show strong inconsistency. Detailed numerical values can be found in Table 2.

the potential of SYNC and our paradigm for synthesizability estimation and synthesizable SBDD. Our contributions can be summarized as:

- **Comprehensive Benchmarks**. We collect and organize 5 datasets containing both easy- and hard-to-synthesize molecules from various data sources to benchmark 8 classical synthesizability metrics. Additionally, we also benchmark 11 existing SBDD methods using these metrics.

- **A Practical Metric.** We propose a 3D-aware and SE(3)-invariant *SYNthesizability Classifier* (SYNC) as a synthesizability metric, reaching better performance in accuracy and efficiency. Designed with multiple merits, SYNC is consequently suitable for guiding SBDD methods.

- **A Synthesizable Paradigm.** Our proposed SYNC can be directly combined with existing SBDD methods with two controllable generation methods - guided diffusion and DPO - to improve the synthesizability of the generated molecules while not sacrificing their original binding affinity.

## 2 RELATED WORK

### 2.1 STRUCTURE-BASED DRUG DESIGN

Structure-Based Drug Design has evolved tremendously in recent years, driven by generative AI models. Some early works iteratively generate atom voxelization density maps, occupied grids, or atom types/positions in an autoregressive manner, including 3DSBDD (Luo et al., 2021), and Pocket2Mol (Peng et al., 2022). Later research paradigms shifted to diffusion-based models, including TargetDiff (Guan et al., 2023), DiffBP (Lin et al., 2025), and GraphBP (Liu et al., 2022), which enable generation of full-atom positions and atom types simultaneously. Functional group-based methods, FLAG (Zhang et al., 2023) and D3FG (Lin et al., 2024a), have incorporated prior knowledge on fragment motifs into model design. DecompDiff (Guan et al., 2024) and its subsequent works, DecompOPT (Zhou et al., 2024) and DecompDPO (Cheng et al., 2024), break molecules into arms and scaffolds, design each arm more carefully and connect them with scaffolds to generate new molecules. Recent works, such as MolCraft (Qu et al., 2024) and VoxBind (Pinheiro et al., 2024), adopt latest and more powerful molecular denoising techniques, achieving promising generation results. For more methods, we refer readers to a recent benchmark (Lin et al., 2024b).

## 2.2 SYNTHESIZABILITY EVALUATION

Existing works can be broadly categorized into three branches: (1) Rule-based methods, such as SA (Ertl & Schuffenhauer, 2009) and SYBA (Voršilák et al., 2020), are based on predefined functional group compatibility, structural stability, etc. (2) Retrosynthetic-based methods, such as Aizynthfinder (Genheden et al., 2020), aim to decompose a given molecule and recursively search for purchasable precursors. (3) Learning-based approaches, such as Nonpher (Voršilák & Svozil, 2017), RAscore (Thakkar et al., 2021), GASA (Yu et al., 2022), and FSScore (Neeser et al., 2023), learn to flexibly model synthesizable landscapes of the chemical space, which show better generalization. Despite much progress, limitations exist. Rule-based methods are naive and often lead to sub-optimal performance. Retrosynthetic-based methods ensure the highest accuracy for molecules classified as synthesizable by providing synthetic routes, but are incredibly time-consuming. Moreover, the rich information underlying 3D conformation is neglected by learning-based methods.

## 2.3 SYNTHESIZABLE MOLECULE DESIGN

Current dominant approach of generating synthesizable molecules involves chemical building blocks (i.e., accessible molecular fragments) and reaction templates. Early methods explored Bayesian optimization (Korovina et al., 2020), genetic algorithms (Gao et al., 2021), and Monte Carlo tree search (Swanson et al., 2024). Recently, GFlowNet-based (Bengio et al., 2023) methods (Shen et al., 2023; Koziarski et al., 2024; Cretu et al., 2024; Seo et al., 2024) have emerged with great potential. However, these models **do not generate 3D molecule structures** and cost extra computation for docking 2D molecules into 3D protein pockets, which draws a clear boundary with our method. Although CGFlow (Shen et al., 2025) generates 3D conformations as an exception, it does not guarantee structural quality. We provide additional illustrations and comparisons with these building block-based methods in **Appendix A**. Synthesizability in 3D molecule design is yet to be improved.

## 3 PRELIMINARIES

**Structure-Based Drug Design** SBDD aims to generate 3D ligands that bind to a given protein pocket with high binding affinity. The protein pocket and 3D ligand can be represented as $P = \{(x_P^{(i)}, v_P^{(i)}, e_P^{(i,j)})\}_{i,j=1}^{N_P}$ and $M = \{(x_M^{(i)}, v_M^{(i)}, e_M^{(i,j)})\}_{i,j=1}^{N_M}$, where $N_P, N_M$ denotes the number of atoms in the protein pocket and ligand, respectively, and $x \in \mathbb{R}^3$, $v \in \mathbb{R}^d$, and $e \in \mathbb{R}^{N \times N}$ denote the coordinates of atoms, the types of atoms, and the types of bonds between atoms.

**Molecule Generation via Diffusion** Based on the Denoising Diffusion Probabilistic Model (DDPM) framework (Ho et al., 2020), most diffusion models for SBDD can be formulated as a conditional Markov chain by the (forward) diffusion process that gradually adds noise to data:

$$q(M_{1:T} \mid M_0, P) = \prod_{t=1}^{T} q(M_t \mid M_{t-1}, P). \tag{1}$$

The generative process denoises to generate molecules with a neural network parameterized by $\theta$:

$$p_\theta(M_{0:T-1} \mid M_T, P) = \prod_{t=1}^{T} p_\theta(M_{t-1} \mid M_t, P), \tag{2}$$

where $M_1, M_2, \cdots, M_T$ is a sequence of latent variables with the same dimensionality as data $M_0 \sim p(M_0 \mid P)$, and $p(M_T)$ is a prior data distribution during the sampling process.

## 4 BENCHMARKING SYNTHESIZABILITY METRICS

We first study the rationality of existing synthesizability metrics within a benchmark before moving into synthesizable molecule design. The central challenge in establishing such a benchmark is to collect molecules that can and cannot be synthesized. Synthesizable molecules are readily available, e.g., any molecule that can be purchased from a commercial library is synthesizable. However, it is tricky to determine that a molecule cannot be synthesized without the aid of expensive experimental assays. Therefore, we take easy- and hard-to-synthesize molecules as a cheap but good alternative, because it is easier to collect hard-to-synthesize molecules than completely unsynthesizable ones.

**Datasets** Five datasets are used as benchmark datasets. Firstly, we utilize Enamine building block stock (Enamine, 2023) as the ES dataset, ensuring that all molecules are readily purchasable. Moreover, we collect three test sets (TS1/TS2/TS3) (Yu et al., 2022) and the Nonpher-Test dataset from (Voršilák & Svozil, 2017), which contain both ES and HS molecules labeled by different previous

Table 1: **Classification accuracies** for synthesizability metrics across 5 datasets, where the best and second metrics on each dataset are marked as **bold** and underline, respectively (same for all subsequent tables). For scoring metrics (SA, SYBA, RAScore, SCScore), we apply thresholds to distinguish easy- and hard-to-synthesize molecules. For FSScore, since no record for the threshold has been found, we dynamically adjust its threshold to guarantee the accumulative best performance over five datasets. All metrics are categorized as retrosynthetic-based, rule-based, and learning-based approaches, labeled using •, ⋆, and ▲. More details of thresholds can be found in Appendix C.

| Datasets / Accuracy | Aizynth• | SA⋆ | SYBA⋆ | Nonpher▲ | FSScore▲ | RAScore▲ | SCScore▲ | GASA▲ | SYNC▲ (ours) |
|---|---|---|---|---|---|---|---|---|---|
| TS1 | 0.9535 | 0.9853 | 0.9881 | 0.8817 | 0.5003 | 0.9683 | 0.6518 | 0.9850 | **0.9911** |
| TS2 | 0.7509 | 0.8090 | 0.7369 | 0.7737 | 0.5401 | 0.7136 | 0.4026 | 0.8010 | **0.8406** |
| TS3 | 0.7511 | 0.5667 | 0.5800 | 0.6828 | 0.4717 | 0.6772 | 0.4761 | **0.7590** | 0.7564 |
| Nonpher-Test | 0.5688 | 0.8313 | 0.8812 | 0.8940 | 0.7937 | 0.9250 | 0.4125 | 0.9188 | **0.9313** |
| Enamine | 0.8061 | 0.9844 | 0.9571 | 0.7924 | 0.9171 | **0.9890** | 0.9171 | 0.9538 | 0.9520 |
| *Avg. Rank* | 6.0 | 4.0 | 4.4 | 5.6 | 7.8 | 4.0 | 8.0 | 3.0 | **2.0** |
| Time (s/10K mols) | 174192.60 | 2.67 | 1.70 | **0.97** | 38.99 | 56.81 | 17.4 | 24.93 | 1.78 |

works, and we remove repetitive samples. Note SYNC is trained on an isolated dataset containing millions of 3D molecules excluding test molecules, where details can be found in Appendix B.

A comparison of 9 synthesizability metrics (including our SYNC, detailed in the next subsection) on the 5 datasets is reported in Table 1, from which we can make three important observations:

• **Observation 1**: Rule-based methods often fall short on specific datasets, e.g., SA, SYBA on TS3, demonstrating their limitations in terms of generalization capability. The retrosynthetic-based model can guarantee the highest true positive rate by assuring a synthetic path, but it can still lead to false negatives, since AizynthFinder⋆ fails to find all synthesizable molecules in Enamine. In contrast, learning-based models generally enjoy higher performance but can suffer from a high computational burden, while our work (SYNC) makes the best trade-off between efficiency and effectiveness.

• **Observation 2**: SYNC surprisingly outperforms other baselines by a wide margin in the overall rankings across 5 datasets. For example, SYNC performs best on TS1, TS2, and Nonpher-Test datasets and ranks second on the TS3 dataset, demonstrating its superiority. Although SYNC does not perform best on the Enamine dataset, the gap is not large. This gain is from two aspects: deep learning endows SYNC with higher flexibility in classifying different molecular patterns than rule-based models, while 3D information provides richer information to surpass conventional learning-based methods. For example, *bond angles* and *steric hindrance* are two key factors determining if a molecule can be synthesized, which can only be reflected in the 3D space. We provide a detailed explanation of why 3D is important and investigate the interpretability of SYNC in **Appendix D**.

• **Observation 3**: SYNC achieves excellent performance within limited time consumption. Although Nonpher and SYBA are faster than SYNC, they are significantly less accurate. Both learning-based and retrosynthetic-based methods are computationally heavy; e.g., AizynthFinder, designed for searching retrosynthetic pathways, is unaffordable for large-scale molecule screening.

## 5 METHODOLOGY

### 5.1 SE(3)-INVARIANT SYNTHESIZABILITY CLASSIFIER

In this subsection, we introduce how to train an SE(3)-invariant classifier as the synthesizability metric. Conventional synthesizability predictors often neglect the 3D conformation of target molecules, so rich information is discarded. For example, some out-of-distribution bond angles can significantly increase the synthetic difficulty, or the synthesized molecules can be unstable due to such uncommon structures (Schulman & Venanzi, 1974). Another example is steric hindrance, where atoms or groups within a molecule are brought too close together in space, resulting in unstable molecules (Lenoir et al., 2006; Rösel et al., 2017). More importantly, SBDD methods usually generate 3D conformations and require a 3D-aware synthesizability metric to take conformations into account. Therefore, enabling 3D conformation awareness not only brings additional information that helps predict synthesizability but also makes it a plug-and-play 3D-aware classifier for advancing synthesizable molecular design within our classifier-driven paradigm.

---

⋆Although AizynthFinder is widely used and ensures high accuracy on molecules classified as synthesizable by providing synthetic routes, its overall performance is not optimal. Enlarging the molecular stock (search space) and computational resources may improve results, while we use the official stock for fair comparison.

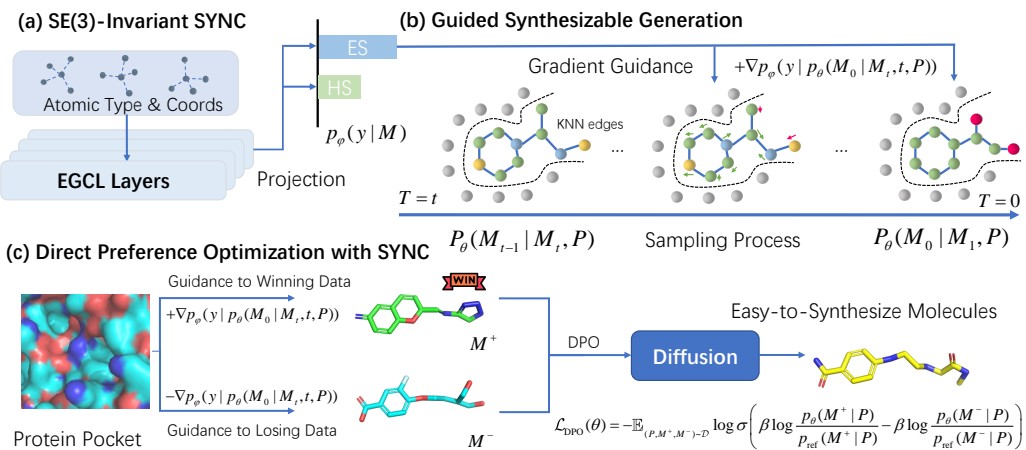

Figure 2: A high-level overview of the Synthesizability Classifier-Driven paradigm. We propose SYNC, an SE(3)-invariant synthesizability classifier, to accurately and efficiently distinguish easy-to-synthesize (ES) and hard-to-synthesize (HS) molecules. We then integrate it as a plug-and-play module with guided diffusion and direct preference optimization (DPO) for SBDD methods.

Based on the above observations and discussion, we believe that a good classifier should have the following advantages: *fast*, *differentiable*, *3D structure-aware*, and *SE(3)-invariant*. Firstly, a good classifier must not only be highly accurate, but also infer fast so that it can be used for large-scale screening to meet practical needs. Secondly, differentiability is another elegant property that benefits the integration of classifier into our synthesizable SBDD paradigm. Thirdly, as mentioned, 3D awareness provides richer information. Finally, synthesizability is intuitively supposed to be agnostic to translations and rotations, so the classifier is expected to be SE(3)-invariant. To meet the above four properties, we propose a simple yet effective SE(3)-invariant *SYNthesizability Classifier* (SYNC) to achieve synthesizability estimation. Assuming we have any arbitrary SE(3)-transformation $T_g$, the prediction output of SYNC is expected to be invariant to $T_g$, as follows:

$$\text{SYNC}(T_g(x_M, v_M, e_M)) = \text{SYNC}(x_M, v_M, e_M). \tag{3}$$

To achieve SE(3)-invariance, we use a $L$-layer EGNN (Satorras et al., 2021) as the backbone of our classifier, where each equivariant Graph Convolutional Layer (EGCL) takes $x_M^{(l)}, v_M^{(l)}, e_M$ as inputs:

$$x_M^{(l+1)}, v_M^{(l+1)} = \text{EGCL}[x_M^{(l)}, v_M^{(l)}, e_M], \tag{4}$$

where $0 \leq l \leq L$, and $x_M^{(0)} = x_M$ and $v_M^{(0)} = v_M$ are the input features. Finally, we directly utilize a nonlinear transformation from $x_M^{(L)}$ to predict synthesizability. Note that we have not designed the classifier to be very complicated, due to the priority of inference efficiency. However, thanks to the merits mentioned above, SYNC is the only choice for our classifier-driven SBDD paradigm.

## 5.2 CLASSIFIER-DRIVEN SBDD PARADIGM 1: GUIDED DIFFUSION

Next, we integrate SYNC with guided diffusion, as shown in Figure 2. The denoising process $p_\theta(M_{t-1} \mid M_t, P)$ in Eq. (2) can be represented as the product of a **Gaussian** distribution $\mathcal{N}$ of continuous atomic coordinates $x_{t-1}$ and a **Categorical** distribution $\mathcal{C}$ of discrete atomic types $v_{t-1}$:

$$p_\theta(M_{t-1} \mid M_t, P) = \mathcal{N}\Big(x_{t-1}; v_\theta([x_t, v_t], t, P), \beta_t I\Big) \times \mathcal{C}\Big(v_{t-1}; c_\theta([x_t, v_t], t, P)\Big), \tag{5}$$

where $\beta_t$ is the simplified variance schedule factor, and $v_\theta([x_t, v_t], t, P)$ and $c_\theta([x_t, v_t], t, P)$ are the predicted data distribution at time $t-1$, where $v_\theta$ and $c_\theta$ can be regarded sub-modules of $p_\theta$ that predict positions and atom types, respectively. We next add guidance from a pre-trained classifier (SYNC) $p_\phi(y|M_t)$ to constrain the generated molecules with higher synthesizability, as follows:

$$p_{\theta,\phi}(M_{t-1} \mid M_t, P, y) = p_\theta(M_{t-1} \mid M_t, P)p_\phi(y|M_t)/Z, \tag{6}$$

where $Z$ is a normalizing constant. However, the conditional score $\nabla \log p_\phi(y|M_t)$ is hard to be precisely evaluated because the classifier is trained only on ground-truth molecular conformations

and has not seen intermediate diffusion results. Therefore, we follow (Song et al., 2023; Wu et al., 2024) to approximate $\nabla p_\phi(y|M_t)$ through the estimated output of the denoising network:

$$
\nabla p_\phi(y \mid M_t) = \nabla \int p_\phi(y|M_0)p_\theta(M_0|M_t)\,\mathrm{d}M_0 \approx \nabla \int p_\phi(y|M_0)\delta_{p_\theta(M_0|M_t,t,P)}(M_t)\mathrm{d}M_0
$$
$$
= \nabla p_\phi\Big(y \mid p_\theta(M_0|M_t,t,P)\Big). \tag{7}
$$

For the continuous atom coordinate distribution, the positions for the next timestep are sampled from:

$$
x_{t-1} \leftarrow \mathcal{N}\Big(v_\theta([x_t,v_t],t,P) + \lambda_x \nabla_{x_t} p_\phi\big(y \mid p_\theta(M_0|M_t,t,P)\big), \beta_t I\Big). \tag{8}
$$

As for the categorical atom type distribution, we apply Gumbel softmax to sample from categorical probabilities, and then feed molecules to SYNC and get categorical guidance:

$$
v_{t-1} \leftarrow \mathcal{C}\Big( \exp\Big( \log\big(c_\theta([x_t,v_t],P)\big) + \lambda_v \log \nabla_{v_t} p_\phi\big(y \mid p_\theta(M_0|M_t,t,P)\big)\Big)\Big), \tag{9}
$$

where $\lambda_x$ and $\lambda_v$ control the strength of guidance in terms of atomic coordinates and types.

**Multi-Step Guidance**   As the diffusion step $t$ approaches 0, the model may suffer from *diffusion confinement*, i.e., the model will stubbornly keep atoms in some particular spatial patterns. To address this issue, we introduce multi-step guidance, which allows for multiple guidance steps in a single diffusion step. The reason for not simply setting larger factor $\lambda$ but utilizing multi-step guidance is that larger $\lambda_x$ tends to move the atoms too far away from each other. As a result, the bonds between atoms no longer exist, and the molecule is torn into pieces, forming isolated atoms.

**Edge Construction**   Another challenge is that the diffusion process does not involve molecule bonds. Some previous works (Guan et al., 2024; Peng et al., 2023) integrate edge diffusion so that at each timestamp $t$, the ligand is no longer a set of atom points, but the complete molecule. In this paper, we address this issue in another straightforward way. When training SYNC, we utilize the ground-truth bonds but ignore their bond type information, which is proven to be harmless to performance in Section 6.5. During guidance, we connect atoms by finding their $K$-nearest neighbors based on the maximum valency of each atom. We also filter bonds longer than 2.0 Å and make sure all bonds are bidirectional; otherwise, atoms with higher valency may mistakenly connect to other atoms. Such a design can solve the atom-bond inconsistency during diffusion generation.

### 5.3   Classifier-Driven SBDD Paradigm 2: Direct Preference Optimization

After introducing training-free controllable diffusion, we explore how to fine-tune the generative model with SYNC by DPO (Rafailov et al., 2024) in Figure 2. In practice, how to obtain high-quality pairwise preference data is a central challenge for DPO. However, generating data directly using diffusion models does not work because most of the generated molecules are hard-to-synthesize. Different from previous methods (Zhou et al., 2024) that use the classifier directly as a reward function in the DPO framework, we utilize SYNC to **generate better pairwise preference data** in this paper. Specifically, we follow the strategy described in Section 5.2 to generate easy- and hard-to-synthesize molecules $(M^+, M^-)$ either by adding or subtracting the gradients in Eqs. (8)(9).

The vanilla DPO framework (Rafailov et al., 2024) fine-tunes the model on pairwise preference data $(M^+, M^-)$:

$$
\mathcal{L}_{\mathrm{DPO}}(\theta) = -\mathbb{E}_{(P,M^+,M^-)\sim\mathcal{D}} \log \sigma \left( \beta \log \frac{p_\theta(M^+|P)}{p_{\mathrm{ref}}(M^+|P)} - \beta \log \frac{p_\theta(M^-|P)}{p_{\mathrm{ref}}(M^-|P)} \right), \tag{10}
$$

where $M^+$ is the preferred (easy-to-synthesize) molecule, $M^-$ is the unfavored (hard-to-synthesize) molecule, and $\beta$ is a factor controlling DPO strength. In addition, $p_\theta(\cdot|\cdot)$ is the target model to be fine-tuned, and $p_{\mathrm{ref}}(\cdot|\cdot)$ is the frozen original model to constrain the tuned model not too far away. Following (Wallace et al., 2024), the DPO loss in diffusion can be formulated as follows:

$$
\mathcal{L}_{\mathrm{Diffusion\text{-}DPO}}(\theta) = -\mathbb{E}_{(P,M^+,M^-)\sim\mathcal{D}} \log \sigma\Big(\beta\mathbb{E}_{M_{1:T}^+,M_{1:T}^-}\Big[ \log \frac{p_\theta(M_{0:T}^+ \mid P)}{p_{\mathrm{ref}}(M_{0:T}^+ \mid P)} - \log \frac{p_\theta(M_{0:T}^- \mid P)}{p_{\mathrm{ref}}(M_{0:T}^- \mid P)}\Big]\Big). \tag{11}
$$

Using Jensen's inequality and the convexity of $\log \sigma$, the following inequality can be derived:

$$
\mathcal{L}_{\mathrm{Diffusion\text{-}DPO}}(\theta) \leq -\mathbb{E}_{(P,M^+,M^-)\sim\mathcal{D},t\sim U(0,T)} \log \sigma
$$
$$
\Big(\beta\Big[ \log \frac{p_\theta(M_{t-1}^+|M_t^+,P)}{p_{\mathrm{ref}}(M_{t-1}^+|M_t^+,P)} - \log \frac{p_\theta(M_{t-1}^-|M_t^-,P)}{p_{\mathrm{ref}}(M_{t-1}^-|M_t^-,P)}\Big]\Big). \tag{12}
$$

Based on Eq. (5), the above loss is further simplified as:

$$\mathcal{L}_\theta = -\mathbb{E}_{(P,M^+,M^-)\sim\mathcal{D},t\sim U(0,T)} \log \sigma\Big(-\beta T\omega(\lambda_t)\Big(\mathcal{L}_D\big(M^+, p_\theta(M_t^+, t, P)\big)-$$

$$\mathcal{L}_D\big(M^+, p_{\text{ref}}(M_t^+, t, P)\big) - \big(\mathcal{L}_D(M^-, p_\theta(M_t^-, t, P)) - \mathcal{L}_D(M^-, p_{\text{ref}}(M_t^-, t, P))\big)\Big)\Big), \quad (13)$$

where $\mathcal{L}_D$ is the original diffusion loss, $\lambda_t$ is the signal-to-noise ratio, $\omega(\lambda_t)$ is a weighting function (usually a constant in practice (Ho et al., 2020)), and we factor the constant $T$ into $\beta$. The final loss $\mathcal{L}_\theta$ in Eq. (13) encourages the model to improve generative possibility at sample $M^+$ than sample $M^-$, which finally leads to higher overall synthesizability of generated molecules.

# 6 EXPERIMENT

## 6.1 EXPERIMENTAL SETUP

**Datasets, Baselines, Metrics** We follow previous works (Luo et al., 2021) to use the Cross-Docked2020 (Francoeur et al., 2020) dataset for training and evaluating SBDD methods. We sample 100 molecules for each protein pocket in the test set, resulting in a total of 10,000 molecules. For pairwise data for DPO, to prevent data leakage, we randomly sample 10,000 pairwise preference data points from the training set and filter a total of 3,431 pairs satisfying the ES and HS molecules annotated by SYNC for TargetDiff, and 1,991 pairs for DecompDiff. We consider multiple classical synthesizability metrics, including AizynthFinder, FSScore, GASA, SA, SYBA, SCScore, RAScore, Nonpher, and SYNC. To benchmark synthesizability, we consider 3DSBDD, LiGAN, D3FG, DecompDiff, DiffBP, FLAG, GraphBP, MolCraft, Pocket2Mol, TargetDiff, and VoxBind.

**Motivation for Backbones** As a milestone in SBDD, TargetDiff is often applied as a backbone model in later works (Huang et al., 2024; Qian et al., 2024), and we follow this setting. While methods like FLAG and GraphBP excel in synthesizability, they lag in binding affinity. We believe that focusing solely on synthesizability while neglecting binding affinity is unwise. Conversely, DecompDiff performs well in terms of binding affinity but lacks synthesizability, so we intend to overcome this dilemma to generate molecules with high binding affinity and synthesizability.

## 6.2 CLASSIFIER-DRIVEN PARADIGM GENERATED SYNTHESIZABLE MOLECULES

Table 2: **Synthesizability ratings** of generated drugs for various SBDD methods. ($\uparrow$)/($\downarrow$) denotes that a **larger/smaller** value is better. The Top 2 results are marked by **bold** and underline. Since the optimal direction varies across metrics (i.e., larger-is-better vs. smaller-is-better), for clarity, we uniformly use *green* and *red* to denote *improved* and *reduced* synthesizability w.r.t the backbones.

| Methods | Aizynth• ($\uparrow$) | SA★ ($\uparrow$) | SYBA ($\uparrow$) | Nonph. ($\uparrow$) | FSSc. ($\uparrow$) | RASc. ($\uparrow$) | SCSc.($\downarrow$) | GASA ($\uparrow$) | SYNC ($\uparrow$) |
|---|---|---|---|---|---|---|---|---|---|
| 3DSBDD | 0.1960 | 0.635 | -25.795 | 0.2538 | -4.310 | 0.538 | 3.202 | 0.3321 | 0.2978 |
| D3FG | 0.0728 | 0.564 | -56.213 | 0.1599 | -8.097 | 0.407 | 3.679 | 0.2061 | 0.1041 |
| DiffBP | 0.0968 | 0.609 | -27.891 | 0.2556 | -5.671 | 0.446 | 3.364 | 0.2118 | 0.2533 |
| FLAG | 0.0760 | 0.582 | -29.799 | 0.3608 | **-0.181** | 0.759 | **2.466** | **0.5639** | 0.2345 |
| GraphBP | 0.1936 | **0.690** | -8.862 | 0.3988 | -3.843 | **0.794** | 3.032 | 0.5538 | 0.2180 |
| LiGAN | **0.2280** | 0.663 | **-2.471** | **0.4538** | -2.413 | 0.605 | 3.027 | 0.4302 | 0.3348 |
| MolCraft | 0.1929 | 0.672 | -14.381 | 0.3909 | -5.938 | 0.533 | 3.495 | 0.3935 | 0.3292 |
| Pocket2Mol | 0.1389 | 0.654 | -27.329 | 0.3529 | -2.368 | 0.683 | 2.899 | 0.4168 | 0.1630 |
| VoxBind | 0.1296 | 0.657 | -24.362 | 0.3333 | -6.388 | 0.570 | 3.500 | 0.3442 | 0.2365 |
| TargetDiff | 0.1124 | 0.601 | -41.991 | 0.2328 | -6.560 | 0.421 | 3.522 | 0.2418 | 0.1958 |
| TargetDiff-Guide | 0.1365 | 0.626 | -33.123 | 0.2793 | -5.270 | 0.427 | 3.423 | 0.2653 | **0.3977** |
| $\Delta_{\text{TargetDiff}}$ | +21.4% | +4.2% | +21.1% | +20.0% | +19.7% | +1.4% | -2.8% | +9.7% | +100.3% |
| TargetDiff-DPO | 0.1283 | 0.626 | -34.396 | 0.2844 | -5.437 | 0.455 | 3.417 | 0.2683 | 0.2385 |
| $\Delta_{\text{TargetDiff}}$ | +14.1% | +4.2% | +18.1% | +22.2% | +18.5% | +8.1% | -3.0% | +11.0% | +21.8% |
| DecompDiff | 0.0176 | 0.574 | -16.006 | 0.2739 | -12.860 | 0.330 | 4.414 | 0.2104 | 0.1201 |
| DecompDiff-Guide | 0.0263 | 0.584 | -8.788 | 0.3108 | -12.227 | 0.337 | 4.359 | 0.2239 | 0.1378 |
| $\Delta_{\text{DecompDiff}}$ | +49.4% | +1.7% | +45.1% | +13.5% | +5.0% | +2.1% | -1.2% | +6.5% | +14.7% |
| DecompDiff-DPO | 0.0241 | 0.586 | -14.940 | 0.3105 | -11.349 | 0.300 | 4.211 | 0.1762 | 0.1529 |
| $\Delta_{\text{DecompDiff}}$ | +36.9% | +2.1% | +6.7% | +13.4% | +11.8% | -9.1% | -4.6% | -16.3% | +27.31% |

We first benchmark in Table 2 the molecular synthesizability of 11 classical SBDD methods, from which some interesting observations can be made. For example, D3FG and DecompDiff, which were previously shown to perform well on binding due to the consideration of fragmental and decomposable priors, have relatively poor synthesizability. Instead, some earliest methods, such as FLAG and GraphBP, perform well in terms of synthesizability despite their simplicity.

Further, we demonstrate the effectiveness of **TargetDiff/DecompDiff-Guide** and **-DPO**, two means of the synthesizability classifier-driven SBDD paradigm, in terms of molecular synthesizability using two SBDD methods as the backbone model. While there are significant inconsistencies across various synthesizability metrics, as analyzed in Figure 1, when SYNC is integrated into our classifier-driven SBDD paradigm, nearly all synthesizability metrics are consistently improved. It can be found that TargetDiff-Guide and TargetDiff-DPO both outperform vanilla TargetDiff on all 9 metrics, and TargetDiff-Guide achieves the best performance on SYNC. Besides, TargetDiff-Guide performs much better than TargetDiff-DPO on the SYNC metric, since it is directly guided by gradients from SYNC. However, the synthesizability of TargetDiff-DPO still benefits from fine-tuning, as DPO implicitly regulates the model to generate more favorable molecules. More importantly, TargetDiff-DPO enjoys better inference efficiency because it no longer requires tedious and complex calculations of gradients as TargetDiff-Guide does. DecompDiff-Guide also achieves universal synthesizability gains on 9 metrics, while DecompDiff-DPO falls short only on RAScore and GASA. The performance gain is relatively lower than TargetDiff, which we hypothesize is caused by the inner molecular complexity of DecompDiff since it generates larger molecules and is naturally less synthesizable than TargetDiff. Note that we only take these two SBDD methods as a case to show the great potential of our synthesizability classifier-driven paradigm, and do not claim any brand-new generative models. More discussion on properties of generated molecules is in Appendix E.

## 6.3 BINDING AFFINITY COMPARISON

Table 3: Comparison of binding affinity for various SBDD methods. Performance gains and drops are marked in *green* and *red*, respectively.

| Methods | Vina Score ($\downarrow$) | | Vina Min ($\downarrow$) | | Vina Dock ($\downarrow$) | |
|---|---|---|---|---|---|---|
| | Avg. | Med. | Avg. | Med. | Avg. | Med. |
| 3DSBDD | - | - | -3.75 | - | -6.45 | - |
| D3FG | - | - | -2.59 | - | -6.78 | - |
| DiffBP | - | - | - | - | -7.34 | - |
| FLAG | - | - | - | - | -3.65 | - |
| GraphBP | - | - | - | - | -4.80 | -4.70 |
| LiGAN | - | - | - | - | -6.33 | -6.20 |
| MolCraft | -6.61 | **-8.14** | **-8.14** | **-8.42** | **-9.25** | **-9.20** |
| Pocket2Mol | -5.14 | -4.70 | -6.42 | -5.82 | -7.15 | -6.79 |
| VoxBind | **-6.94** | -7.11 | -7.54 | -7.55 | -8.30 | -8.41 |
| TargetDiff | -5.33 | -6.08 | -6.47 | -6.60 | -7.42 | -7.68 |
| TargetDiff-Guide | -5.63 | -6.39 | -6.76 | -6.85 | -7.75 | -7.83 |
| $\Delta_{\text{TargetDiff}}$ | -0.30 | -0.31 | -0.29 | -0.25 | -0.33 | -0.15 |
| TargetDiff-DPO | -5.40 | -6.06 | -6.51 | -6.55 | -7.58 | -7.66 |
| $\Delta_{\text{TargetDiff}}$ | -0.07 | +0.02 | -0.04 | +0.05 | -0.16 | +0.02 |
| DecompDiff | -3.50 | -5.80 | -6.57 | -7.22 | -8.39 | -8.43 |
| DecompDiff-Guide | -3.64 | -5.84 | -6.54 | -7.45 | -8.60 | -8.44 |
| $\Delta_{\text{DecompDiff}}$ | -0.14 | -0.04 | +0.03 | -0.23 | -0.21 | -0.01 |
| DecompDiff-DPO | -3.48 | -5.73 | -6.44 | -7.34 | -8.57 | -8.68 |
| $\Delta_{\text{DecompDiff}}$ | +0.02 | +0.07 | +0.13 | -0.12 | -0.18 | -0.25 |

We have controlled the generation of molecules in the synthesizable chemical space, but have not imposed any restrictions on the binding of ligands to the pocket. Therefore, we further explore whether our paradigm would harm the performance of vanilla backbone models in binding-related metrics. As shown in Table 3, we find that generated molecules enjoy similar binding affinity within limited fluctuation when compared to the vanilla backbone models. This suggests that our paradigm not only significantly improves synthesizability, but also preserves or even enhances binding affinity in particular situations at the same time. Synthesizable molecules usually adopt stable conformations that better complement protein pockets. In contrast, unfavorable molecules often contain unrealistic substructures, such as distortions or oversized rings, which hinder pocket binding.

## 6.4 CASE STUDY ON GENERATED MOLECULES

We provide a case of molecules generated by TargetDiff, TargetDiff-Guide, and TargetDiff-DPO, along with an analogy projected by ChemProjector (Luo et al., 2024) from an HS molecule generated by TargetDiff, and we report their binding affinities. We can observe that TargetDiff-Guide and TargetDiff-DPO enhance the binding (docking) strength to the protein pocket, despite not utilizing any additional binding constraints. Another advantage lies in the fact that our methods improve synthesizability mainly through minor modifications to local substructures of the original molecules, which is quite different from previous works, such as ChemProjector, which projects each molecular fragment onto purchasable building blocks, leading to more severe structural changes. Moreover, our methods can directly generate ES molecules with excellent binding affinity and synthesizability in one single step and do not require extra 2D-to-3D mapping, ranking of top molecules, and multiple times of molecule docking. For more details and case studies, please refer to Appendix F.

## 6.5 ABLATION STUDIES

**Edge Reconstruction** As mentioned in Section 5.2, we ignore edge features to mitigate inconsistencies from neglecting bonds during diffusion. Theoretically, single, double, and triple bonds can be implicitly captured by neighboring atoms, while $\pi$ bonds are preserved by assigning distinct tokens to atoms on $\pi$ rings. To test this design, we compare SYNC with SYNC-Edge. Table 4 shows only a minor gap, suggesting that modeling edge features is not decisive for synthesizability.

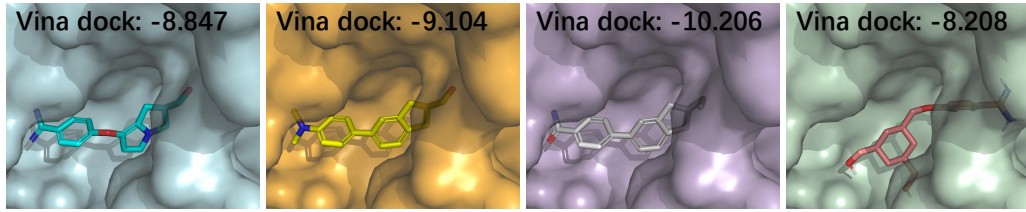

| (a) TargetDiff (HS) | (b) TargetDiff-Guide (ES) | (c) TargetDiff-DPO (ES) | (d) ChemProjector (ES) |

Figure 3: Examples when TargetDiff is applied as the backbone on PDB 5LIU. All ES molecules are double-verified by SYNC and AizynthFinder. We first input the HS molecule in (a) to ChemProjector, which generates a 2D analog molecule. Then we generate its 3D conformation and dock it to the protein pocket by AutoDock Vina to get the conformation and binding affinity in (d). The synthesizability gain is not derived from correcting basic errors in HS molecular structures, but rather from the rich knowledge in SYNC that enables the modification of atomic types and coordinates.

**3D Conformation**    Most SBDD methods generate full-atom types and coordinates simultaneously, making a SE(3)-invariant SYNC essential. Beyond our default 3D SYNC, we also test variants using 1D fingerprints (FPS), 1D SMILES, and 2D graphs. As shown in Table 4, 3D SYNC ranks first on three datasets and second on one, highlighting the importance of 3D information for synthesizability.

Table 4: Performance comparison of whether or not to consider edge features and information of different modalities.

| Methods | TS1 | TS2 | TS3 | Nonpher-Test | Enamine |
|---|---|---|---|---|---|
| **SYNC (3D)** | 0.9911 | **0.8406** | 0.7564 | **0.9313** | **0.9520** |
| **SYNC-Edge** | 0.9846 | 0.8391 | 0.7575 | 0.9250 | 0.9366 |
| **SYNC-1D-FPS** | 0.9899 | 0.8081 | 0.7422 | 0.8250 | 0.9370 |
| **SYNC-1D-SMILES** | **0.9947** | 0.8335 | **0.7917** | 0.7250 | 0.9321 |
| **SYNC-2D-Graphs** | 0.9879 | 0.8332 | 0.7722 | 0.9188 | 0.9473 |

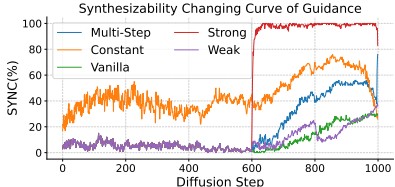

Figure 4: Curves of synthesizability with diffusion steps utilizing different guidance strategies on PDB 1WN6.

**Guidance Settings**    As discussed in Section 5.2, we apply SYNC guidance to SBDD methods in a multi-step manner. Here, we provide an analysis to validate its effectiveness. Figure 4 compares four guidance settings against vanilla TargetDiff by plotting the percentage of synthesizable molecules over diffusion steps. Our main setting, *Multi-Step*, begins guidance in the latter half of training (from step 600). Although synthesizability improves, it often drops in the final steps; applying multiple guidance steps at this stage produces a sharp increase. Adjusting guidance strength (*Strong* or *Weak*, controlled by $\lambda_x$ and $\lambda_v$) yields very different outcomes: *Strong* guidance boosts synthesizability but breaks molecular structures into fragments (Appendix G), while *Weak* guidance has little effect. The *Constant* setting, where guidance is applied throughout, performs worse than the vanilla version, suggesting early guidance is ineffective or even harmful since molecules are not yet well-formed.

**DPO Settings**    To examine the effect of DPO, we track the percentage of synthesizable molecules using SYNC and the SA score Ertl & Schuffenhauer (2009) during fine-tuning. Both metrics rise steadily in the first 2,000 iterations, peaking at iteration 1,800, which we finally select as the optimal checkpoint. Further training, however, leads to performance fluctuation and generates fractured molecules, most likely due to overfitting on the pairwise preference dataset. Recall that we have sampled 3,431 ES-HS pairs and use a batch size of 4, the peak performance corresponds to roughly two training epochs.

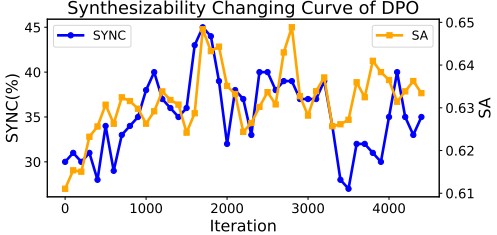

Figure 5: Curve of synthesizability (SYNC percentage and SA Score) with DPO iterations on the protein (pdb: 1WN6), when TargetDiff is applied as the backbone model as a showcase.

## 7 CONCLUSION

In this work, we first provide a comprehensive benchmark on synthesizability metrics, showing their drawbacks and the lack of consistency. We then propose a simple yet effective _SYNthesizability Classifier_ (SYNC), which performs the best on the benchmark datasets. With SYNC as a plug-and-play module, we construct a synthesizability classifier-driven SBDD paradigm, with two classic SBDD methods, TargetDiff and DecompDiff, to generate highly synthesizable molecules while preserving their binding affinity. Extensive experiments demonstrate that our methods can effectively improve chemical properties and synthesizability metrics on the generated molecules. Even though promising results show our potential to meet more practical needs of the pharmaceutical industry, we acknowledge this paper is an exploratory work for improving synthesizability with respect to _de novo_ designed molecules. We leave limitations such as wet-lab confirmation for future work.

### ACKNOWLEDGEMENTS

This work was supported by National Science and Technology Major Project (No.2022ZD0115101), National Natural Science Foundation of China Project (No.624B2115 No.U21A20427), the National Science and Technology Major Project of China (No.2021YFA1301603), the Project (No. WU2025B006) from the SOE Dean Special Project Fund (SOE-DSPF) Program of Westlake University, Project (No. WU2022A009) from the Center of Synthetic Biology and Integrated Bioengineering of Westlake University and the Hangzhou Postdoctoral Daily Funding Program (No.103140026582502, 2025).

### REPRODUCIBILITY STATEMENT

We have taken several steps to ensure the reproducibility of our results. First, all applied datasets are open-sourced and can be directly downloaded. Second, we have appended a GitHub link in the abstract, which contains source codes for SYNC, TargetDiff/DecompDiff-Guide/DPO. Third, we conduct detailed ablation studies to show hyperparameters. Finally, we describe all datasets and experiment details in Appendix B.

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

## USE OF LLM

We only apply LLM for checking spelling and grammar.

## A    COMPARISONS WITH GFLOWNET-BASED METHODS

GFlowNet-based (Bengio et al., 2023) methods (Shen et al., 2023; Koziarski et al., 2024; Cretu et al., 2024; Seo et al., 2024) have emerged to show great potential in generating synthesizable molecules. However, these methods do not produce 3D conformations, which are critical in structure-based drug design. CGFlow (Shen et al., 2025) stands out as it applies GFlowNet for synthesizable molecule generation and generates 3D conformation when designing molecules for the target protein pocket. Therefore, we take it as a representative work of this series. However, as shown in Figure A1, the generated molecules clash with the protein pocket, distorting their shapes. We set the protein surface to be partially transparent in order to visualize the regions where clashes occur with the protein, i.e., the parts of the molecule shown in lighter colors in the figure. This demonstrates that, despite the tremendous success of GFlowNet-based and building block-based methods in designing synthesizable molecules, their ability to design ligands for protein pockets remains limited. Conversely, our method fills this gap by modifying the type and coordinate of each atom to improve synthesizability.

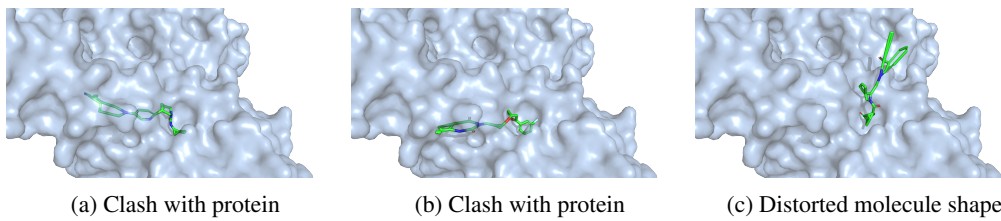

(a) Clash with protein          (b) Clash with protein          (c) Distorted molecule shape

Figure A1: Examples of ligands generated by CGFlow on 1K9T protein target.

## B    DATASET & PLATFORM DESCRIPTION

We benchmark a total of 9 synthesizability metrics on 5 datasets. We denote easy- and hard-to-synthesize molecules as ES and HS for simplification in the following. Here are the detailed descriptions of these involved datasets:

**TS1**    TS1 is directly obtained from SYBA (Voršilák et al., 2020), which contains 3,581 ES molecules randomly selected from the ZINC15 database and 3,581 HS molecules picked from the GDB-17 (Ruddigkeit et al., 2012) database.

**TS2**    TS2 is obtained from ChEMBL, GDBChEMBL, and GDBMedChem, which is used as the test set in RAscore (Thakkar et al., 2021) and displays a broader distribution in the chemical space (Thakkar et al., 2021). TS2 contains 17,348 ES molecules and 13,000 HS molecules.

**TS3**    TS3 is composed of 900 ES and 900 HS molecules, which are collected from various sources and are more challenging to distinguish, including (1) 203 ES molecules selected from the ZINC15 database used by SYBA (Voršilák et al., 2020); (2) 40 HS molecules derived from the evaluation of 296 published compounds by experienced medicinal chemists (Huang et al., 2011; Ertl & Schuffenhauer, 2009; Boda et al., 2007; Fukunishi et al., 2014); (3) 518 molecules collected from the Molecule Libraries Small Molecule Repository (MLSMR) from PubChem (Kim et al., 2016); (4) 1039 molecules from  (Sheridan et al., 2014) whose complexity is evaluated by chemists.

**Nonpher Test**    The compounds in the Nonpher-Test dataset are obtained by analyzing 296 published structures, whose ease of synthesis was assessed by experienced medicinal chemists. Data sources are described in (Voršilák & Svozil, 2017). It contains 120 ES and 40 HS molecules.

**Enamine**  We utilize molecules from the Enamine US Stock catalog, retrieved in October 2023 (Enamine, 2023), which contains 211,220 molecules. Since all molecules are purchasable, they are all categorized as ES molecules.

**Train Set**  First, small molecules with an SAscore between 3.5 and 6 from the ChEMBL (Gaulton et al., 2017) and GDBChEMBL (Bühlmann & Reymond, 2020) databases are collected. In SAscore, although the authors recommend 6 as the classification threshold, it can be found from several studies that the synthesizability of molecules distributed between 3.5 and 6 is quite ambiguous (a large number of HS compounds with an SAscore less than 6), and thus are closer to the decision boundary (Gao & Coley, 2020; Thakkar et al., 2021). Then, the potential synthetic routes for the molecules were predicted by Retro* (Chen et al., 2020) (a multistep retrosynthetic planning algorithm that predicts the synthetic routes for target molecules). The molecules that require less than 10 synthetic steps are marked as ES, and those that need more than 10 reaction steps or cannot be successfully predicted by Retro* are labeled as HS. Second, except for the sampling through retrosynthetic analysis, the data sets used in SYBA (Voršilák et al., 2020) are also used in our study. The positive examples were collected from the purchasable compounds in the ZINC15 (Sterling & Irwin, 2015) database, and the negative examples were generated by Nonpher (Voršilák & Svozil, 2017) (a molecular morphing algorithm for the generation of HS structures). Pairs of compounds (ES and HS) with fingerprint similarities higher than 0.35 are selected to improve the model's sensitivity in distinguishing similar structures. Finally, the data set consisting of a total of 800,000 compounds was randomly split into the training, validation, and test set by a ratio of 8:1:1 based on stratified sampling. Note that these works are done in (Yu et al., 2022); we just follow their settings.

**Platform**  We train and evaluate our SYNC on a single NVIDIA A100-80G GPU. We also use A100 to complete TargetDiff/DecompDiff-Guidance/DPO experiments. It takes about 20 minutes to sample a single pocket using TargetDiff, and 50 minutes using DecompDiff.

## C  THRESHOLDS FOR SCORING METRICS

The thresholds for each metric are listed in Table A1. Notably, we dynamically adjusted the threshold with a granularity of 1 for FSScore, since no reference value is reported to our best knowledge.

Table A1: Thresholds of each metric.

| SA | SYBA | FSScore | RAScore | SCScore |
|------|-------|---------|---------|---------|
| 0.61 | -18.6 | -13 | 0.5 | 3.1 |

We visualize the overall distributions of metrics to show the sensitivity of each threshold in Figure A2. Due to space and time limitations, we visualize the distributions of SA, SYBA, RAScore on TS1. Based on these figures, the threshold is within an appropriate margin.

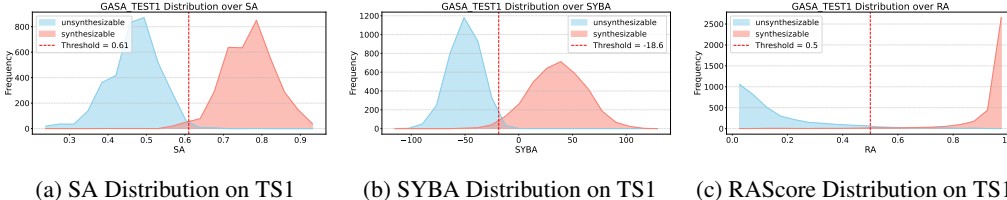

(a) SA Distribution on TS1    (b) SYBA Distribution on TS1    (c) RAScore Distribution on TS1

Figure A2: Threshold and score distribution on TS1.

However, there is no reference value for FSScore. From the visualizations in Figure A3 on different datasets, FSScore cannot distinguish molecules perfectly, so we dynamically adjust the threshold, so the cumulative performance on 6 datasets is maximized. We can observe that FSScore fails to distinguish molecules in most scenarios.

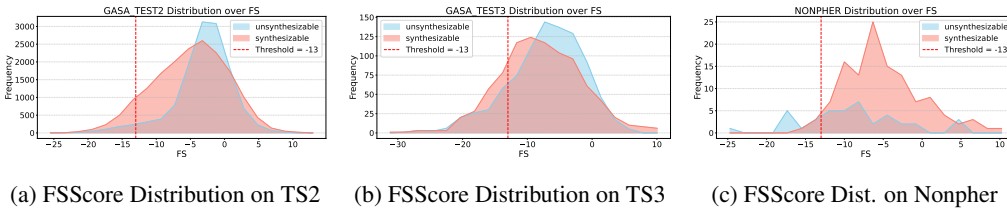

(a) FSScore Distribution on TS2    (b) FSScore Distribution on TS3    (c) FSScore Dist. on Nonpher

Figure A3: Distribution of FSScore on different datasets.

## D INTERPRETABILITY OF SYNC

**3D Conformation** Conformation can provide rich extra information for judging whether a molecule is synthesizable. Molecules with extreme bond angles are typically more challenging to synthesize. For example, tetrahedrane (Schulman & Venanzi, 1974) (C12C3C1C23) remains unsynthesized to date due to its 60° bond angles, which significantly hinder the laboratory preparation of tetrahedrane. Another example is steric hindrance, where atoms or groups within a molecule are brought too close together in space, resulting in unstable molecules. For example, tetra-tert-butylethylene (Lenoir et al., 2006) remains unsynthesized to date due to its overcrowded spatial distribution. We visualize their 3D conformation in Figure A4. As shown in Table A2, when tested on all synthetic metrics, only AizynthFinder and SYNC can properly classify these molecules as hard-to-synthesize molecules, whereas all other methods fail to give a correct answer. AizynthFinder predicts correctly because these molecules have no precursors, whereas SYNC benefits from its unique 3D structural view. However, all other methods fall short due to the neglect of 3D information.

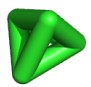

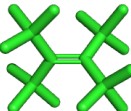

(a) Tetrahedrane. (Extreme bond angles.)    (b) Tetra-tert-butylethylene. (Steric hindrance.)

Figure A4: Examples of two hard-to-synthesize molecules, due to their uncommon 3D structures.

Table A2: Predictions made by synthetic metrics. ✓ stands for prediction is correct (predicted as hard-to-synthesize), and × stands for prediction is incorrect (predicted as easy-to-synthesize).

| Molecule | Aizynth[•] | SA[⋆] | SYBA | Nonph. | FSSc. | RASc. | SCSc. | GASA | SYNC |
|---|---|---|---|---|---|---|---|---|---|
| Tetrahedrane | ✓ | × | × | × | × | × | × | × | ✓ |
| Tetra-tert-butylethylene | ✓ | × | × | ✓ | × | × | × | × | ✓ |

**Gradient Visualization** To further investigate what structural pattern can influence the judgment of SYNC, we visualize the gradients of molecules from three sources in Figure A5. From red to blue, the importance of each atom is in descending order. (a) A hard-to-synthesize molecule from TS2. Red atoms are in the center of this molecule, indicating that SYNC does not prefer overlapping rings. (b) An example from Compas (Wahab et al., 2022), which contains thousands of computational photoelectric molecules and is hard-to-synthesize. This also shows a similar lack of preference for overlapping rings. (c) An SBDD-generated molecule. SYNC significantly dislikes the heptatomic ring structure, which has far less frequency of aromatic rings.

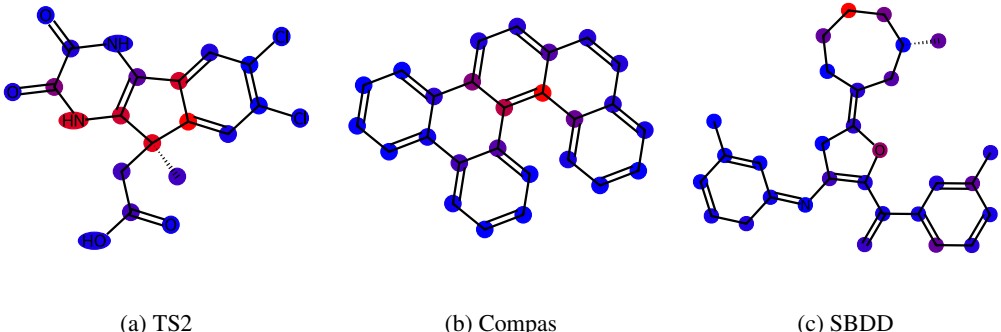

(a) TS2  (b) Compas  (c) SBDD

Figure A5: Three hard-to-synthesize samples and the gradients backpropagated by SYNC.

# E  PROPERTY OF GENERATED MOLECULES

Following previous works (Guan et al., 2023; 2024), we discuss drug-likeness(QED (Bickerton et al., 2012)), molecule diversity, uniqueness, and novelty in Table A3. It turns out that the diversity of TargetDiff-Guide and TargetDiff-DPO is higher than TargetDiff, showing the superiority of our method. The QED and diversity of DecompDiff-Guide both improved. While QED for DecompDiff-DPO slightly drops, the diversity significantly increases, showing DecompDPO generates a variety of molecules. This conclusion is also reflected by uniqueness and novelty, all the methods generate highly diverse molecules (uniqueness) and not over similar (novelty) to those in the training set.

Table A3: The statistical analysis of generated molecules.

| Methods | QED↑ | Diversity (Mean)↑ | Diversity (Median)↑ | Uniqueness↑ | Novelty↑ |
|---|---|---|---|---|---|
| TargetDiff | 0.487 | 0.744 | 0.738 | **0.978** | 0.981 |
| TargetDiff-Guide | **0.512** | 0.761 | 0.758 | 0.972 | **0.981** |
| TargetDiff-DPO | 0.503 | **0.765** | **0.766** | 0.976 | 0.977 |
| DecompDiff | 0.393 | 0.596 | 0.588 | 1.0000 | 0.995 |
| DecompDiff-Guide | **0.394** | 0.597 | 0.589 | 1.0000 | 0.994 |
| DecompDiff-DPO | 0.377 | **0.652** | **0.645** | 1.0000 | **0.996** |

Table A4: The pass ratio of each test provided by PoseBusters.

| Methods | Bond Lengths | Bond Angles | Steric Clash | Ring Flatness | Double Bond Flatness | Internal Energy |
|---|---|---|---|---|---|---|
| TargetDiff | 0.973 | 0.762 | 0.903 | **1.000** | 0.999 | 0.712 |
| TargetDiff-Guide | 0.977 | 0.734 | 0.941 | 0.999 | **1.000** | 0.695 |
| TargetDiff-DPO | **0.978** | **0.772** | **0.945** | **1.000** | 0.999 | **0.753** |
| DecompDiff | 0.942 | **0.919** | 0.908 | 0.999 | 0.959 | 0.736 |
| DecompDiff-Guide | 0.934 | 0.916 | 0.903 | **1.000** | **0.966** | 0.740 |
| DecompDiff-DPO | **0.953** | 0.885 | **0.947** | 0.999 | 0.937 | **0.789** |

We further apply PoseBusters (Buttenschoen et al., 2024) to validate the structural validity of the generated molecules comprehensively. A total of six tests are applied to evaluate generated molecules, and we report the pass rate of each test. Bond length evaluates whether bonds are too long or too short; bond angle evaluates whether bond angles are falling in proper distributions; steric clash detects molecules intertwined and atoms clashing; ring and double bond flatness assure generate aromatic rings and double bonds are flat; internal energy blocks twisted rings that are energetically unfavorable. As shown in Table A4, the pass ratio of each test is highly related to the performance of backbones. When tested on TargetDiff, the pass rate for each task almost increases, where experiments on DecompDiff show similar results. Despite occasion performance drops, our methods show overall improvements in all scenarios. This is attributed to our method being able to more effectively guide the generative model to produce structurally more reasonable

small molecules, such as making rings and chemical bonds flatter, thereby generating valid molecular structures.

## F  CASE STUDIES

Projection-based methods (Luo et al., 2024; Gao et al., 2024) take a target molecule as input and generate an analog molecule that is guaranteed to be synthesizable for drug discovery. In particular, they decompose the molecule into fragments by breaking chemical bonds according to predefined reaction templates, then identify similar commercially available building blocks to replace these fragments, thereby constructing a new molecule. We use TargetDiff-generated, hard-to-synthesize molecules as examples. It can be observed in Figure A6 that our method can not only generate ES molecules but also molecules with better binding affinity, showing superiority to fragment-based methods.

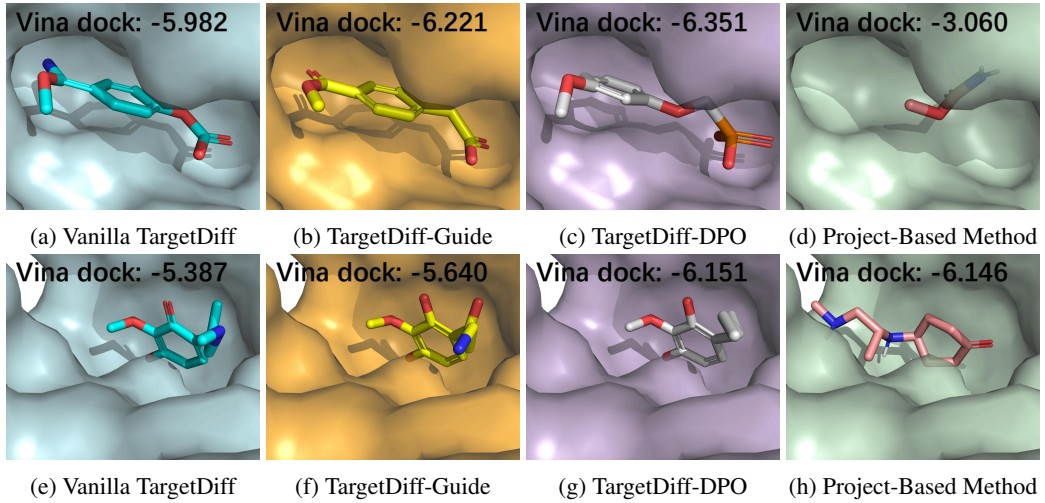

Figure A6: Examples when TargetDiff is applied as the backbone of our classifier-driven paradigm, where the protein targets are 3KC1 (top) and 4F1M (bottom).

## G  GUIDANCE TORN MOLECULES

Here, we present some failure cases in Figure A7 in which molecules are torn into pieces as they receive overly strong guidance signals from SYNC. While some basic functional groups can still be observed, the scaffolds of molecules are completely broken, forming isolated molecule parts.

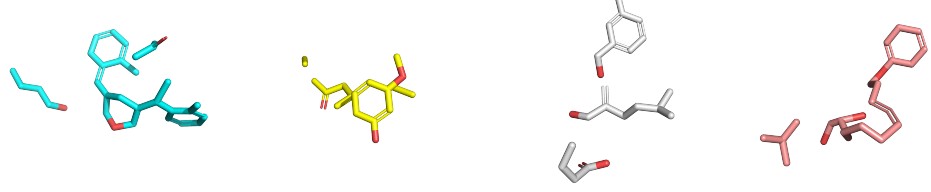

Figure A7: Visualizations of some failure cases, where molecules are generated with overly strong guidance, leading to fatal errors.

