# OpenReview forum: "SYNC: Measuring and Advancing Synthesizability in Structure-Based Drug Design"
_ICLR.cc/2026/Conference — ICLR 2026 Poster_

### Official Review · Reviewer_kwwZ · 2025-10-31

**Soundness:** 3
**Presentation:** 3
**Contribution:** 2
**Rating:** 6
**Confidence:** 3

**Summary:**

The paper introduces SYNC, a 3D-aware, SE(3)-invariant classifier for estimating molecular synthesizability in Structure-Based Drug Design (SBDD).
It benchmarks eight classical synthesizability metrics across 11 SBDD methods, finding strong inconsistencies, and proposes SYNC as a unified, differentiable, and efficient alternative.
SYNC is further embedded into two SBDD paradigms—guided diffusion and Direct Preference Optimization (DPO)—to generate ligands that are both synthesizable and high-affinity.
Extensive experiments (five curated datasets, large-scale SBDD benchmarks) show SYNC achieves higher accuracy and efficiency than existing metrics and improves molecule synthesizability without hurting docking scores

**Strengths:**

This paper makes a timely and practical contribution to structure-based drug design by systematically benchmarking existing synthesizability metrics and introducing a fast, 3D-aware, SE(3)-invariant classifier (SYNC) that effectively bridges molecular generation and realistic synthesis feasibility. The integration of SYNC into both guided diffusion and DPO paradigms is elegant, demonstrating consistent improvements in synthesizability across multiple SBDD backbones without sacrificing binding affinity. The experiments are comprehensive, covering five datasets, eleven baselines, and detailed ablations, making the work both reproducible and impactful for real-world drug discovery.

**Weaknesses:**

While technically sound, the core innovation of SYNC is incremental, primarily reusing standard EGNN architectures for classification. The reliance on proxy “easy/hard-to-synthesize” labels without experimental validation limits the practical reliability of results. Certain methodological aspects, such as gradient guidance stability, parameter sensitivity, and DPO training dynamics, lack clarity. Moreover, evaluations are restricted to two SBDD backbones and omit comparisons with more recent LLM- or synthesis-pathway-based approaches, slightly constraining the generality and novelty of the proposed framework.

**Questions:**

1. How sensitive is SYNC’s performance to the training distribution—does it generalize to unseen chemical scaffolds or pocket types?

2. Can SYNC outputs be calibrated to provide probabilistic synthesizability scores usable in Bayesian optimization?

3. How does SYNC compare with recent LLM-based synthesis predictors (e.g., ChemGPT retrosynthetic models)?

4. Does guided diffusion introduce mode collapse toward overly simple molecules?

5. Are there interpretability tools (e.g., attention maps or feature attribution) to rationalize why SYNC deems a molecule synthesizable?

---

> ### Author Response · Authors · 2025-11-18
> **Rebuttal(1/3)**
>
> Thanks for your insightful and constructive review, we are more than delighted to receive your positive attitude towards our manuscript! We notice you still have several concerns, for which we provide further responses below:
>
> Q1: How sensitive is SYNC’s performance to the training distribution—does it generalize to unseen chemical scaffolds or pocket types?
>
> A1: SYNC’s **training and test sets do not overlap**. As clarified in Appendix B, the training data are derived from ChEMBL, whereas the test sets come from multiple prior works and heterogeneous sources, including ZINC15, PubChem, Enamine, and manually curated collections from chemists. Under this setting, SYNC demonstrates strong generalization across diverse test sets, achieving clear advantages in both accuracy and efficiency.
>
> We additionally conducted a small-scale evaluation on the Compas[1] dataset, which contains many virtually designed optoelectronic molecules that are **generally challenging to synthesize** due to their structural complexity. It contains highly entangled ring-in-ring motifs, resulting in a structurally complex and **relatively homogeneous distribution**. Under this setting, many methods tend to classify nearly all molecules as either ES or HS, causing accuracies to collapse toward 1 or 0. Using a random sample of 1,000 molecules as a simple showcase experiment, SYNC maintains reasonable performance. While this is not intended as a definitive benchmark, the result provides an illustrative indication that **SYNC can generalize beyond the distributions seen during training**.
>
> | Dataset | AiZynthFinder | SA | SYBA | Nonpher | FSScore | RAScore | SCScore | GASA | SYNC |
> | --- | --- | --- | --- | --- | --- | --- | --- | --- | --- |
> | Compas | 1.0 | 0.0 | 0.0 | 1.0 | 0.97 | 0.31 | 0.99 | 1.0 | **1.0** |
>
> Moreover, we added an additional experiment to a **new protein target (8DYG)** using TargetDiff as the backbone, serving as a showcase, which is not included in the CrossDocked2020 dataset. Here is the result:
>
> |  | AiZynthFinder | SA | SYBA | Nonpher | FSScore | RAScore | SCScore(↓) | GASA | SYNC |
> | --- | --- | --- | --- | --- | --- | --- | --- | --- | --- |
> | TargetDiff | 0.1912 | 0.6888 | -16.322 | 0.4118 | -5.585 | 0.6818 | 3.3732 | 0.4559 | 0.2500 |
> | TargetDiff-Guide | 0.2686(+0.0774) | 0.7036(+0.0148) | -9.462(+6.860) | 0.4478(0.0360) | -5.813(-0.228) | 0.7068(+0.0250) | 3.2687(-0.1045) | 0.5075(+0.0516) | 0.2985(+0.0485) |
> | TargetDiff-DPO | 0.2500(+0.0588) | 0.7014(+0.0126) | -10.266(+6.056) | 0.4306(+0.0188) | -5.605(-0.020) | 0.7140(+0.0322) | 3.3623(-0.0109) | 0.4861(+0.0302) | 0.2917(+0.0417) |
>
> SYNC also shows reasonably consistent behavior when applied to different protein targets, **suggesting that it retains a degree of transferability in this setting.**
>
> Q2: Can SYNC outputs be calibrated to provide probabilistic synthesizability scores usable in Bayesian optimization?
>
> A2: Yes. Although the current version of SYNC outputs binary labels, its predictions **can naturally be converted into continuous probability scores** through a softmax function, reflecting the model’s confidence in whether a molecule is ES or HS. In fact, our implementation already uses these softmax-based continuous scores to monitor whether the guidance process is increasing a molecule’s synthesizability, and we consistently observe that the predicted ES probability rises during guidance. These calibrated probabilities can be directly used as objectives or constraints in Bayesian optimization, as they provide a continuous and comparable confidence measure. A higher probability indicates stronger confidence in synthesizability. **Therefore, SYNC’s outputs can be readily extended into a probabilistic form suitable for Bayesian optimization.**

---

> ### Author Response · Authors · 2025-11-18
> **Rebuttal(2/3)**
>
> Q3: How does SYNC compare with recent LLM-based synthesis predictors (e.g., ChemGPT retrosynthetic models)?
>
> A3: The most relevant retrosynthetic model we could identify under the name "ChemGPT" is a link in [chemgpt.app/retro](https://www.chemgpt.app/retro), which currently appears to be inaccessible. Despite multiple attempts, the interface consistently returned network errors, preventing us from performing a direct empirical comparison. So we give some theoretically analysis instead:
>
> SYNC and recent LLM-based synthesis predictors **serve related but distinct purposes**. SYNC is a geometry-aware EGNN classifier focused on assessing *whether* a molecule is synthesizable, using explicit structural inductive bias and requiring comparatively modest training data. In contrast, LLM-based models retrosynthesis systems aim to generate *how* to synthesize a molecule (routes, reactants, steps) from large textual reaction corpora.
>
> Because SYNC is a discriminative, structure-driven predictor, it is **efficient and suitable for large-scale screening** or as a synthesizability filter in design **or optimization, as a plug-and-play module**. LLM-based models provide **richer generative capabilities** but are **typically heavier, slower**, and optimized for route prediction rather than fast binary (or probabilistic) feasibility assessment. The two approaches can be complementary, and SYNC can naturally integrate with LLM-based planners by first filtering molecules and then passing viable candidates to a route-generation model.
>
> Hopefully this analysis can ease your concern, if you have specific preference of “ChemGPT” or require other comparison with specific LLM-based models, please feel free to share with us, we are happy to attend to them!
>
> Q4: Does guided diffusion introduce mode collapse toward overly simple molecules?
>
> A4: We assume that the “overly simple molecules’’ you are referring to are those with fewer than 10 atoms (*this number is just a case study*). Using TargetDiff as an example, we evaluate the validity of each setting:
>
> |  | TargetDiff | TargetDiff-Guide | TargetDiff-DPO |
> | --- | --- | --- | --- |
> | Validity | 0.9796 | 0.9765 | 0.9755 |
>
> As shown, the validity rate (i.e., successful construction and passing the RDKit check) **remains essentially unchanged**. This suggests that **guided diffusion and DPO do not induce mode collapse toward overly simple molecular structures.**
>
> Q5: Are there interpretability tools (e.g., attention maps or feature attribution) to rationalize why SYNC deems a molecule synthesizable?
>
> A5: Yes, we performed this analysis in **Appendix D**, which may have been overlooked. There, we visualize per-atom gradients to assess how each atom contributes to SYNC’s prediction. The red-to-blue color scale reflects decreasing importance. The results indicate that for hard-to-synthesize molecules, **SYNC consistently highlights the structurally challenging regions**, such as entangled ring-in-ring motifs and seven-membered rings.
>
> W1: The core innovation of SYNC is incremental
>
> A6: Instead of proposing a brand new algorithm for SBDD, we focus on addressing the long-standing issue of **insufficient synthesizability in structure-based drug design (SBDD)** by building on existing methodologies. Our main contributions are as follows.
>
> 1. We identify and characterize the strong inconsistencies among current synthesizability evaluation methods, and introduce SYNC—**a simple, fast, and accurate classifier** that incorporates 3D conformational information to assess synthesizability more reliably.
> 2. We conduct a **systematic benchmark** of eight existing synthesizability metrics, providing the community with a rigorous and fair comparison that has not been available previously.
> 3. We develop **a SYNC-guided controllable generation paradigm** based on guidance and DPO, achieving consistent improvements in synthesizability across two different backbones.
>
> Together, these contributions offer a new perspective for the field, and the plug-and-play nature of SYNC provides a practical tool for both evaluating and improving synthesizability.
>
> To summarize, we focus on the reward model of synthesizability of SBDD, including metrics and benchmarks, and then combine reward alignment techniques with specific designs to improve existing methods and successfully validate this paradigm, setting a new standard for SBDD.

---

> ### Author Response · Authors · 2025-11-18
> **Rebuttal(3/3)**
>
> W2: The reliance on proxy “easy/hard-to-synthesize” labels without experimental validation limits the practical reliability of results.
>
> A7: **Your concern is precisely a bottleneck in this research field.** While confirming a molecule as synthesizable is relatively easy, defining a molecule as unsynthesizable is extremely hard unless tried every approach mastered by humans. Therefore, **we adopt the concept of easy/hard-to-synthesize (ES-HS) labeling from previous experts in chemistry**[2-4] to achieve synthesizability estimation to the best of our ability. We believe that with fundamental progress in chemistry, this proxy can be enhanced in the near future.
>
> W3: Certain methodological aspects, such as gradient guidance stability, parameter sensitivity, and DPO training dynamics, lack clarity.
>
> A8: Thanks for your careful feedback! In fact, we have conducted several clarifications which you might want to check.
>
> **Gradient guidance stability:**
> In Section 6.5 *Guidance Settings, Figure 4*, we provide a detailed comparison of several designs. To ensure stable guidance, we adopt the Multi-Step strategy. As discussed in Section 6.5, **Weak** guidance (small λ) yields limited improvement, while **Strong** guidance (large λ) makes molecules collapse into isolated atoms. **Constant** guidance improves synthesizability early on but fails in the final steps because the backbone applies strong late-stage updates that a fixed λ cannot counteract. Our **Multi-Step** schedule gradually increases λ near the end and applies several guidance updates per step, achieving a balance between structural stability and improved synthesizability.
>
> **DPO training dynamics:**
>
> In Section 6.5 *DPO Settings*, we discuss the training details of DPO. One challenge of DPO is its tendency to overfit, making the choice of the stopping point crucial. To avoid letting the backbone overfit to SYNC, we also monitor the SA metric to track synthesizability changes. As shown in Figure 5, both SYNC and SA steadily improve during the first 1800 iterations (~2 epochs), after which further training leads to a decline. We therefore select the checkpoint at 1800 iterations as the final model.
>
> **Parameter sensitivity:**
>
> Aside from these hyperparameters, the remaining ones belong to SYNC itself. In the rebuttal, we added new experiments evaluating SYNC under different depths and widths. The results show that SYNC is not highly sensitive to these hyperparameters, and the 4-layer, 128-dimension configuration performs among the best, which is why we chose it as our final model.
>
> | Hyperparameter | TS1 | TS2 | TS3 | Nonpher-Test | Enamine |
> | --- | --- | --- | --- | --- | --- |
> | 3L 128D | 0.9810 | 0.7995 | 0.7394 | 0.9250 | 0.9027 |
> | 3L 256D | 0.9889 | 0.8347 | 0.7405 | 0.9313 | 0.9402 |
> | **4L 128D** | **0.9911** | 0.8406 | **0.7564** | 0.9313 | 0.9520 |
> | 4L 256D | 0.9892 | 0.8415 | 0.7456 | 0.9438 | 0.9431 |
> | 5L 128D | 0.9901 | 0.8331 | 0.7291 | **0.9625** | 0.9450 |
> | 5L 256D | 0.9908 | 0.8374 | 0.7445 | 0.9188 | **0.9525** |
> | 6L 128D | 0.9901 | **0.8425** | 0.7365 | 0.9250 | 0.9496 |
> | 6L 256D | 0.9680 | 0.7978 | 0.7252 | 0.9313 | 0.8983 |
>
> If you have any remaining concerns about the experimental details, please let us know. We would be glad to clarify them further.
>
> W4: Moreover, evaluations are restricted to two SBDD backbones.
>
> A9: As outlined in the paper, we selected these two backbones with specific motivations in mind. TargetDiff is widely adopted as a backbone in follow-up work, whereas DecompDiff offers strong binding affinity but relatively limited synthesizability. This contrast makes them a suitable pair for examining how to achieve both properties simultaneously. Our goal is to illustrate the feasibility of a classifier-driven SBDD paradigm, showing that it can enhance synthesizability while maintaining binding affinity.
>
> [1]Wahab A et al. The compas project. Journal of Chemical Information and Modeling, 2022, 62(16): 3704-3713.
>
> [2]Voršilák, et al. SYBA. Journal of cheminformatics, 2020.
>
> [3]Neeser et al. FSScore. arxiv preprint 2023.
>
> [4]Yu et al. GASAJournal of Chemical Information and Modeling, 2022
>
> In light of these responses, we hope we have addressed your concerns. If we have left any notable points of concern unaddressed, please do share and we will attend to these points! We sincerely hope that you can appreciate our efforts on responses and revisions and thus raise your score.

---

### Official Review · Reviewer_FJHL · 2025-11-01

**Soundness:** 4
**Presentation:** 4
**Contribution:** 3
**Rating:** 8
**Confidence:** 3

**Summary:**

The authors compared classical synthesizability metrics for classical SBDD methods and observed inconsistencies in the rankings. They thus devised a framework called SYNC as a new metric that achieved more consistent reports on various datasets. Also, since they designed such a framework to be fast, differentiable, and SE-invariant, they demonstrated that such a metric can be incorporated into diffusion guidance and DPO. Experiments showed that such incorporation benefits the synthesizability without hurting much of the affinity. They performed ablation studies to justify their choice for the framework as well as guidance, such as the use of 3D information and multistep guidance.

**Strengths:**

1. The presentation is straightforward and clear.
2. The designed framework and model are well-suited for this task and provide even more benefits, such as speed and robustness.
3. The experiments are strong and support the claims well.

**Weaknesses:**

1. Since SYNC is trained on the same datasets shown in Table 1, it is not entirely fair to call it "surprisingly outperforms other baselines by a wide margin", as it is data-centric. This can be related to Question 1.

2. In Table 2, many color markings are off. Minuses are marked as both green and red. This could potentially lead to false interpretation.

3. SYNC's framework can be better shown with some illustrations. So far, it was only described in plain text and mentioned only halfway down in the paper. If this is one of the main contributions, please be more explicit in the main section.

**Questions:**

1. Why is SYNC so low on the generative models’ results compared to the constructed datasets such as TS1 and TS2? I wonder how much of this drop is due to the limited capacity of the model and how much is caused by its lack of generalization to out-of-distribution samples.

2. Why is there a sudden spike in SYNC during the final steps of the multi-step process? Figure 4 may require additional explanations.

---

> ### Author Response · Authors · 2025-11-18
> **Rebuttal**
>
> Thanks for your insightful and constructive review, we are more than delighted to receive positive feedback on our manuscript! We make several further clarifications as follows:
>
> W1 & Q1: Why is SYNC so low on the generative models’ results compared to the constructed datasets such as TS1 and TS2?  Since SYNC is trained on the same datasets shown in Table 1, it is not entirely fair to call it "surprisingly outperforms other baselines by a wide margin”.
>
> A1: Thanks for your careful consideration, and there seems to be some misunderstanding.
>
> First, SYNC is **not** trained on any data from the test sets. It is trained solely on the ES–HS pairs carefully curated in prior work. This is described in Appendix B, and we will further emphasize it in the main text. In contrast, the test sets consist of small collections of molecules manually gathered by chemists across several previous studies, and therefore do not share a unified data distribution. SYNC has never seen any of these test molecules, yet achieves both high accuracy and fast inference, which we find genuinely surprising.
>
> Second, regarding the results presented in Tables 1 (TS1 & TS2) and 2 (generation results), the evaluation metrics differ in definition.
>
> • In Table 1, the metric is **classification accuracy**, measuring how many ES–HS pairs are correctly classified.
>
> • In Table 2, the metric instead reflects **the synthesizability rating of generated molecules**, i.e., the proportion of ES molecules(classification) or degree (regression) of designed molecules.
>
> These tables, therefore, **capture different quantities**, and there is no contradiction implying that SYNC fails to provide a reasonable estimate for molecules generated by design models due to distribution shift. We appreciate your feedback and highlighted this distinction more clearly in the table notes to prevent potential confusion for readers.
>
> W2: In Table 2, many color markings are off. Minuses are marked as both green and red. This could potentially lead to false interpretation.
>
> Thank you for your thoughtful concern. The apparent inconsistency arises because the evaluation metrics differ in their scoring conventions: for most metrics, higher values correspond to better synthesizability, **whereas a few metrics—such as SCScore—follow the opposite trend.** To avoid confusion, we adopted a visualization style similar to that commonly used in financial reporting, where green denotes an improvement and red denotes a decline in terms of synthesizability. Accordingly, all values marked in green indicate improved performance (i.e., more synthesizable molecules), even in cases where the raw metric values decrease.
>
> We have emphasized this point more clearly in the table header to avoid potential misunderstanding.
>
> W3: SYNC's framework with some illustrations.
>
> A3: Thanks for your constructive advice! We have revised the main figure and decomposed SYNC into EGCL layers[1] and linear layers, with illustrations of inputs and outputs.
>
> Q2: Why is there a sudden spike in SYNC during the final steps of the multi-step process?
>
> A4: Please first consider the *Constant* curve in Figure 4, which corresponds to applying a uniform guidance strength at every timestep. When comparing it with the *Vanilla* curve, we observe a clear drop in synthesizability during the final 100 steps. This suggests that the original backbone performs relatively strong modifications to the molecule near the end of the diffusion process, and the fixed guidance strength in *Constant* is not sufficient to counterbalance these changes.
>
> Next, consider the *Strong* curve, where we simply increase the guidance strength (i.e., use a larger λ). This indeed yields higher synthesizability. However, as illustrated in Appendix G, the guidance becomes overly aggressive and leads to “fragmentation” of the molecules, which is undesirable for our purposes.
>
> Based on these observations, we gradually increase the guidance strength during the final few hundred steps (a mild increase in λ) and introduce additional update steps (multiple times of guidance in a single diffusion step) via the Multi-Step strategy, allowing the molecule more opportunities to adapt toward synthesizable configurations near the end of the diffusion process. In particular, during the **very last steps—where the backbone’s predicted structures are already stabilizing—we further increase the number of Multi-Step iterations to give the molecule sufficient chances to be refined.** This progressive schedule produces the spike observed in the final steps, ensuring that more molecules become synthesizable. The multi-step design allows atoms requiring substantial adjustments to be updated repeatedly, while atoms needing only slight refinements can remain stable. This achieves **a balance between improving synthesizability and preserving structurally reasonable molecules**.
>
> [1] Satorras V G, et al EGNN
>
> In light of these responses, we hope we have addressed your concerns.

---

### Official Review · Reviewer_ejoL · 2025-11-01

**Soundness:** 3
**Presentation:** 3
**Contribution:** 3
**Rating:** 6
**Confidence:** 3

**Summary:**

The paper tackles a central bottleneck in structure-based drug design by focusing on synthesizability. It first assembles five datasets with easy- and hard-to-synthesize labels and benchmarks eight widely used metrics across eleven SBDD generators, showing large and sometimes contradictory judgments among existing metrics. To address this, it introduces SYNC, an SE(3)-invariant, 3D-aware classifier that predicts synthesizability from ligand conformations. SYNC is then used as a plug-and-play control signal in two paradigms, guided diffusion and direct preference optimization, to steer TargetDiff and DecompDiff toward molecules that are easier to make while largely maintaining docking affinity. Extensive experiments, ablations, and runtime measurements suggest SYNC attains strong accuracy with low inference cost and yields consistent gains in synthesizability without notable loss in binding metrics.

**Strengths:**

The strengths lie in the clarity and importance of the problem framing, the breadth of the benchmark, and the empirical finding that many classical synthesizability metrics disagree in practice. The proposed classifier is well motivated for SBDD because it explicitly consumes 3D information and is designed to be SE(3)-invariant and fast, which are practical requirements for generation-time control. The integration pathways are simple yet effective, and results are consistently favorable across multiple metrics and two different backbones, with helpful tables that isolate improvements and show that binding scores remain similar or sometimes improve. The ablations are thoughtful, covering 3D versus 2D or 1D inputs, edge modeling choices, guidance schedules, and DPO iteration behavior, which strengthens the case that the observed gains are not brittle. The work also pays attention to computational cost and scalability, an often overlooked but critical aspect for large-scale screening and iterative design.

Here is the itemized summary for rebuttal and discussion:

- Clearly frames the practical importance of synthesizability in structure-based drug design
- Builds a broad benchmark with multiple datasets, metrics, and generators, revealing contradictions among common metrics
- Introduces a fast SE(3)-invariant, 3D-aware classifier that matches SBDD needs
- Provides two simple integration paths, guidance and DPO, that act as plug-and-play controls
- Improves synthesizability consistently while largely preserving docking-based binding scores

**Weaknesses:**

The weaknesses stem from evaluation design and possible circularity. The ES and HS labels are proxies rather than wet-lab confirmations, so real-world synthesizability remains inferred rather than demonstrated. Several baselines require thresholding or stock assumptions that can bias outcomes, and FSScore’s threshold is tuned over the same datasets, which may inflate apparent fairness. Because SYNC both evaluates and guides generation, parts of the analysis risk becoming self-referential, even if cross-metric reporting partly mitigates this. The DPO pairs are constructed using SYNC’s guidance, which could further entangle training and evaluation signals. The guided diffusion approach introduces extra compute and hyperparameters, and the K-nearest-neighbor edge construction and valency heuristics during guidance, while pragmatic, could be fragile across chemistries or larger rings. Reported binding comparisons rely on docking scores, which are useful but imperfect surrogates; stronger tests such as pose validity checks or physics-based rescoring would raise confidence. Finally, although SYNC often ranks first, margins narrow on some datasets, and the method does not outperform retrosynthetic tools on their strongest cases, leaving open how best to combine fast 3D classifiers with pathway-level verification for end-to-end design.

---

1. Uses proxy labels rather than wet-lab validation, leaving real-world synthesizability unconfirmed
2. Several baselines depend on thresholds and stock assumptions that can bias comparisons, with possible tuning leakage
3. Risk of circularity since SYNC both evaluates and guides, and DPO pairs derive from SYNC-guided samples
4. Guidance adds compute and hyperparameters; KNN edges and valency heuristics may be brittle across chemistries
5. Reliance on docking scores without stronger pose checks or physics-based rescoring limits confidence
6. Advantages narrow on some datasets and do not exceed retrosynthesis tools on their best cases
7. External validity to more diverse targets and chemotypes remains to be demonstrated

**Questions:**

-

---

> ### Author Response · Authors · 2025-11-18
> **Rebuttal (1/2)**
>
> Thanks for your insightful and constructive review, we are more than delighted to receive your positive attitude towards our manuscript! We notice you still have several concerns, for which we provide further responses below:
>
> W1: Uses proxy labels rather than wet-lab validation, leaving real-world synthesizability unconfirmed
>
> A1: **Your concern is precisely a bottleneck in this research field.** While confirming that a molecule *can* be synthesized is relatively straightforward, declaring a molecule *unsynthesizable* is inherently difficult—this would require exhausting essentially all synthetic strategies currently known to humans. To approximate this challenging distinction, we **follow prior work in chemistry** and adopt the “easy-to-synthesize / hard-to-synthesize” (ES–HS) labeling paradigm[1–3], which provides the most practical proxy for synthesizability available at present. We expect that as fundamental advances in synthetic chemistry continue, this proxy will become increasingly accurate and informative.
>
> W2: Several baselines depend on thresholds and stock assumptions that can bias comparisons, with possible tuning leakage.
>
> A2: Thank you for the thoughtful feedback. We have taken care to ensure that all methods are compared under fair and consistent conditions.
>
> **For threshold-based approaches**, we followed established practice by adopting boundary values reported in prior studies. The only exception is FSScore, for which no precedent exists. As illustrated in Figure A3 of Appendix C, FSScore exhibits limited intrinsic ability to distinguish between synthesizable and non-synthesizable molecules; therefore, we applied a dynamic threshold. Even with this adjustment, its performance (i.e., the best case) remains comparatively weak.
>
> **For stock-based methods**, expanding the stock size without constraint could bias the comparison. To avoid such unfair advantages, all evaluations were conducted using the officially released stock.
>
> W3: Risk of circularity since SYNC both evaluates and guides, and DPO pairs derive from SYNC-guided samples
>
> A3: We understand and appreciate this concern. Indeed, using SYNC to evaluate generations optimized by SYNC introduces the possibility of circularity. However, as you pointed out, despite SYNC being involved in both guidance and evaluation, we observe **consistent and substantial gains** in synthesizability **across eight additional and diverse external metrics**. This robustness strongly suggests that the improvements are genuine rather than artifacts of self-preference.
>
> If SYNC were simply overfitting to its own scoring pattern rather than enhancing true synthesizability, such improvements would not persist **across two backbone models (TargetDiff and DecompDiff), four optimization strategies (Guidance and DPO), and nine synthesizability metrics in total**. The breadth and consistency of these results indicate that the method generalizes beyond SYNC itself.
>
> W4: Guidance adds compute and hyperparameters; KNN edges and valency heuristics may be brittle across chemistries
>
> A4: (1) Guidance adds affordable computational overhead while providing meaningful improvements in synthesizability. **To further reduce the inference burden**, we introduce DPO, which preserves the inference speed of the vanilla backbones while still producing more synthesizable molecules.
>
> (2) Guidance introduces two hyperparameters: **guidance strength $\lambda$ and guidance schedule**. For both hyperparameters, we have provided detailed ablations in Figure 4, Sec. 6.5. We show our guidance settings achieve good performance among several possible hyperparameter combinations.
>
> (3) We also considered the potential brittleness of the KNN construction. Table 4 shows that removing edge information and adopting a purely KNN-based heuristic (SYNC-3D) does not degrade performance. Across five diverse test sets, SYNC-3D performs competitively with SYNC-Edge, indicating that **this simplification is both stable and reasonable.**
>
> W5: Reliance on docking scores without stronger pose checks or physics-based rescoring limits confidence
>
> A5: We appreciate your careful consideration. We are aware of this point as well, and **in Table A4 of Appendix E** we have conducted a more fine-grained analysis of molecular–protein binding using **PoseBusters**[4], which evaluates whether generated complex conformations satisfy basic physicochemical criteria across multiple metrics. The results show that our method maintains a pass rate highly similar to that of the original backbone. Given that we only introduced constraints on synthesizability, this finding indicates that **our approach not only preserves docking scores but also retains reasonable molecular–protein binding patterns and physicochemical properties.**

---

> ### Author Response · Authors · 2025-11-18
> **Rebuttal(2/2)**
>
> W6: Advantages narrow on some datasets and do not exceed retrosynthesis tools on their best cases
>
> A6: When considering the overall performance and cumulative accuracy across the five datasets, **SYNC clearly surpasses all baselines**, indicating that its performance is genuinely competitive. It is also worth noting that SYNC achieves this level of accuracy while maintaining very fast inference speed, making it a practical candidate for large-scale molecular screening.
>
> |  | AiZynthFinder | SA | SYBA | Nonpher | FSScore | RAScore | SCScore | GASA | SYNC |
> | --- | --- | --- | --- | --- | --- | --- | --- | --- | --- |
> | Mean Accuracy | 0.7661 | 0.8353 | 0.8287 | 0.8049 | 0.6446 | 0.8546 | 0.5720 | 0.8835 | **0.8943** |
>
> Regarding your point about SYNC not outperforming retrosynthesis-based methods, the only retrosynthesis method included in our comparison is AiZynthFinder. Under the standard stock setting, SYNC achieves significantly higher accuracy than AiZynthFinder across all five datasets. If your concern pertains to other aspects, we would be happy to address them in detail.
>
> W7: External validity to more diverse targets and chemotypes remains to be demonstrated
>
> A7: We additionally conducted a small-scale evaluation on the Compas[5] dataset, which contains many virtually designed optoelectronic molecules that are **generally challenging to synthesize** due to their structural complexity. It contains highly entangled ring-in-ring motifs, resulting in a structurally complex and *relatively homogeneous distribution*. Under this setting, many methods tend to classify nearly all molecules as either ES or HS, causing accuracies to collapse toward 1 or 0. Using a random sample of 1,000 molecules as a simple showcase experiment, SYNC maintains reasonable performance. While this is not intended as a definitive benchmark, the result provides an illustrative indication that **SYNC can generalize beyond the distributions seen during training**.
>
> | Dataset | AiZynthFinder | SA | SYBA | Nonpher | FSScore | RAScore | SCScore | GASA | SYNC |
> | --- | --- | --- | --- | --- | --- | --- | --- | --- | --- |
> | Compas | 1.0 | 0.0 | 0.0 | 1.0 | 0.97 | 0.31 | 0.99 | 1.0 | **1.0** |
>
> Moreover, we added an additional experiment to **a new protein target (8DYG)** using TargetDiff as the backbone, serving as a showcase, which is not included in the CrossDocked2020 dataset. Here is the result:
>
> Moreover, we added an additional experiment to **a new protein target (8DYG)** using TargetDiff as the backbone, serving as a showcase, which is not included in the CrossDocked2020 dataset. Here is the result:
>
> |  | AiZynthFinder | SA | SYBA | Nonpher | FSScore | RAScore | SCScore(↓) | GASA | SYNC |
> | --- | --- | --- | --- | --- | --- | --- | --- | --- | --- |
> | TargetDiff | 0.1912 | 0.6888 | -16.322 | 0.4118 | -5.585 | 0.6818 | 3.3732 | 0.4559 | 0.2500 |
> | TargetDiff-Guide | 0.2686(+0.0774) | 0.7036(+0.0148) | -9.462(+6.860) | 0.4478(0.0360) | -5.813(-0.228) | 0.7068(+0.0250) | 3.2687(-0.1045) | 0.5075(+0.0516) | 0.2985(+0.0485) |
> | TargetDiff-DPO | 0.2500(+0.0588) | 0.7014(+0.0126) | -10.266(+6.056) | 0.4306(+0.0188) | -5.605(-0.020) | 0.7140(+0.0322) | 3.3623(-0.0109) | 0.4861(+0.0302) | 0.2917(+0.0417) |
>
> SYNC also shows reasonably consistent behavior when applied to different protein targets, **suggesting that it retains a degree of transferability in this setting**.
>
> [1]Voršilák, et al. SYBA. Journal of cheminformatics, 2020.
>
> [2]Neeser et al. FSScore. arxiv preprint 2023.
>
> [3]Yu et al. GASAJournal of Chemical Information and Modeling, 2022
>
> [4]Buttenschoen M et al. PoseBusters. Chemical Science, 2024, 15(9): 3130-3139.
>
> [5]Wahab A et al. The compas project. Journal of Chemical Information and Modeling, 2022, 62(16): 3704-3713.
>
> In light of these responses, we hope we have addressed your concerns. If we have left any notable points of concern unaddressed, please do share and we will attend to these points! We sincerely hope that you can appreciate our efforts on responses and revisions and thus raise your score.

---

### Author Response · Authors · 2025-11-30
**Summary for Rebuttal (1/2)**

Dear (S)ACs,

In light of the recent rebuttal rollback due to information leakage, we have prepared a concise summary of our paper, the reviewers' comments, and our responses. We hope this consolidation helps you quickly understand and evaluate our submission.

This paper focuses on improving the synthesizability of Structure-Based Drug Design (SBDD), which is a significant bottleneck between *in silico* design and wet-lab verification. We propose SYNC to enable faster, more accurate evaluation of synthesizability than existing metrics,  and combine it with classical SBDD methods to achieve synthesizable molecule design without sacrificing binding affinity.

Our main contributions are as follows.

1. We identify and characterize the strong inconsistencies among current synthesizability evaluation methods, and introduce SYNC—**a simple, fast, and accurate classifier** that incorporates 3D conformational information to assess synthesizability more reliably.
2. We conduct a **systematic benchmark** of eight existing synthesizability metrics, providing the community with a rigorous and fair comparison that has not been available previously.
3. We develop **a SYNC-guided controllable generation paradigm** based on guidance and DPO, achieving consistent improvements in synthesizability across two different backbones.

Together, these contributions offer a new perspective for the field, and the plug-and-play nature of SYNC provides a practical tool for both evaluating and improving synthesizability.

Although **we received positive feedback from all reviewers initially**, we would like to highlight specific concerns raised and our corresponding responses. While the rebuttal rollback prevented further dialogue, we have still done our utmost to address every concern raised. We have **faithfully** summarized the key issues below for your reference.

We sincerely appreciate your patience and dedication, especially given the sudden surge in workload. We deeply respect the effort required to maintain such high attention to detail for every submission during this challenging time.

Best,

Authors

---

### Author Response · Authors · 2025-11-30
**Summary for Rebuttal (2/2)**

### Reviewer ejoL

- **On Evaluation Validity (W1, W3):**
    - **Proxy Labels:** We justified using the standard ES–HS paradigm as a necessary proxy, as it is also inherited from previous work.
    - **Circularity:** We disproved circularity concerns by demonstrating consistent gains across **8 external metrics** (independent of our scoring function) and multiple backbones/strategies.
- **On Baselines & Performance (W2, W6):**
    - **Fair Comparison:** We ensured fair comparisons using standard stocks and thresholds, following those from previous works.
    - **SOTA Performance:** SYNC achieves the highest cumulative accuracy across 5 datasets, significantly outperforming retrosynthesis-based methods (e.g., AiZynthFinder) while maintaining fast inference.
- **On Chemical Quality & Complexity (W4, W5):**
    - **Physical Validity:** Using **PoseBusters**, we verified that our method improves synthesizability *without* compromising molecular–protein binding or physicochemical validity.
    - **Efficiency:** We addressed computational concerns by introducing DPO (preserving inference speed) and confirming the stability of our KNN heuristics through ablations.
- **On Generalization (W7):**
    - **New Experiments:** We demonstrated strong generalization on a structurally complex optoelectronic dataset (**Compas**) and an unseen protein target (**8DYG**), proving the method works beyond the training distribution.

### Reviewer **FJHL**

- **On Fairness and Metrics (W1, Q1):**
    - **Risk of Data Leakage:** We clarified that SYNC is trained *solely* on curated pairs from prior work and has never seen the five test sets, ensuring the performance gap is genuine and not due to unfair training.
    - **Metric Distinction:** We resolved the misunderstanding regarding results in Tables 1 and 2: Table 1 reports *classification accuracy* on fixed datasets, while Table 2 reports the *synthesizability score* (proportion/value) of generated molecules. These represent different evaluation dimensions.
- **On Visualization (W2, W3):**
    - **Table Clarity:** We explained that table coloring indicates *improvement* in synthesizability (e.g., lower is better for SCScore), not just numerical increase.
    - **Architecture:** We updated the main figure to clarify the EGCL and linear layers.
- **On Methodological Design (Q2):**
    - **Guidance Schedule:** We justified the "spike" in the final diffusion steps. Our ablation shows that while constant guidance is insufficient and overly strong guidance causes fragmentation, our **adaptive Multi-Step strategy** (increasing guidance intensity only in late stages) successfully refines synthesizability while preserving molecular structural integrity.

### Reviewer **kwwZ**

- **On Generalization & Applicability (Q1, W4):**
    - **Unseen Distributions:** We demonstrated SYNC's robustness on a physically distinct dataset (**Compas**, optoelectronic molecules) and a new protein target (**8DYG**), proving it generalizes beyond training data.
    - **Backbone Selection:** We clarified that the two selected backbones (TargetDiff and DecompDiff) represent distinct regimes (standard baseline vs. high-affinity/low-synthesizability), effectively validating our method's versatility.
- **On Methodological Scope & Comparison (W1, Q3):**
    - **Contribution:** We clarified that our work establishes a **comprehensive benchmark** and a **3D-aware reward model** for SBDD, addressing the "synthesizability alignment" gap rather than just proposing a sampling algorithm.
    - **Vs. LLMs:** We provided a theoretical comparison explaining that SYNC is a fast, structure-based discriminator (suitable for screening/guidance), which is complementary to heavy, generative LLM-based route planners. These analyses are conducted given that the required baseline, “ChemGPT,” is not actually available.
- **On Technical Robustness (Q2, Q4, Q5, W2, W3):**
    - **Stability:** We provided additional ablations on **hyperparameters** (model depth/width) and **training dynamics** (DPO stopping criteria, Multi-step guidance), confirming the method is stable and not brittle.
    - **Probabilistic Output:** We confirmed SYNC can output calibrated probabilities via softmax, making it compatible with Bayesian Optimization.
    - **No Mode Collapse:** We verified that validity rates remain stable under guidance, proving the model does not collapse to overly simple molecules.
    - **Proxy Labels:** We justified using the standard ES–HS paradigm as a necessary proxy, as it is also inherited from previous work.
    - **Interpretability:** We pointed to existing atom-level gradient visualizations (Appendix D) that correctly highlight difficult motifs (e.g., bridged rings).

---

### Meta-Review · Area_Chair_3z1G · 2025-12-26

**Summary:**

Reviewers found the work solid within its chosen setup, with broad benchmarking, clear ablations, and consistent synthesizability gains when using SYNC for guidance and DPO without major drops in docking-based affinity. However, the main concern is that synthesizability is evaluated via ES/HS proxy labels rather than route-level or wet-lab confirmation, leaving real-world feasibility uncertain. Reviewers also flagged residual circularity since SYNC helps both optimize and evaluate generations (including SYNC-shaped DPO pairs), along with sensitivities in baseline thresholding/stock assumptions and limited evidence on generalization. Overall, I lean toward acceptance as a poster, but my confidence is limited.

**Reviewer Concerns:**

The rebuttal addresses several methodological concerns by clarifying that Table 1 (ES/HS accuracy) and Table 2 (synthesizability of generated molecules) measure different quantities, and by providing a concrete justification for the multi-step guidance schedule and its stability. It also strengthens the binding-side evaluation beyond docking by adding PoseBusters-style checks, and offers limited additional evidence of transfer via small showcase experiments.

Key concerns remain. Synthesizability is still grounded in ES/HS proxy labels rather than route-level or experimental validation, leaving real-world feasibility uncertain. There is also residual circularity because SYNC steers generation and shapes DPO pairs while remaining central to evaluation, and broader validation against modern pathway/LLM retrosynthesis tools and more diverse targets is still limited.

**Reviewer Scores:**

- ejoL (6 → 6): Most replies (circularity mitigation via cross-metric gains, added PoseBusters checks, guidance/DPO clarifications) help, but the core limitation they emphasized—proxy ES/HS labels without stronger real-world/route-level validation—still stands, so I expect their score would remain essentially unchanged.

 - FJHL (8 → 8 or 9): Their main issues were largely clarifications (train/test misunderstanding, Table 1 vs 2 interpretation, guidance spike explanation) plus presentation. Since these were directly addressed, I think they would likely stay at 8 and might nudge up to 9 if they felt the added clarifications/figures fully resolved their concerns.

 - kwwZ (6 → 6): The rebuttal adds some extra generalization evidence and explains guidance/DPO dynamics, but their bigger concerns about proxy labels, incremental novelty, and missing strong comparisons to modern route-based/LLM synthesis tools are only partially addressed. I expect they would stay around 6.

---

### Decision · Program_Chairs · 2026-01-26

Accept (Poster)